# CELLPAINTR: CONTRASTIVE BATCH CORRECTION TRANSFORMER FOR LARGE SCALE CELL PAINTING

## ABSTRACT

Cell Painting, a high-content imaging-based profiling method, has emerged as a powerful tool for understanding cellular phenotypes and drug responses. However, batch effects severely constrain the integration and interpretation of data collected across different laboratories and experimental conditions. To mitigate this issue, here we introduce *CellPainTR*, a novel embedding approach through Transformer for unified batch correction and representation learning of Cell Painting data, thereby addressing a critical challenge in the field of image-based profiling. Our approach employs a Transformer-like architecture with Hyena operators, positional encoding via morphological-feature-embedding, and a special source context token for batch correction, combined with a multi-stage training process that incorporates masked token prediction and supervised contrastive learning. Experiments on the JUMP Cell Painting dataset demonstrate that CellPainTR significantly outperforms existing approaches such as Combat and Harmony across multiple evaluation metrics, while maintaining strong biological information retention as evidenced by improved clustering metrics and qualitative UMAP visualizations. Moreover, our method effectively reduces the feature space from thousands of dimensions to just 256, addressing the curse of dimensionality while maintaining high performance. These advancements enable more robust integration of multi-source Cell Painting data, potentially accelerating progress in drug discovery and cellular biology research.

## 1 INTRODUCTION

Cell Painting, a powerful high-content imaging technique, has emerged as a promising tool for biological research and drug discovery. This method generates rich, multidimensional data capturing intricate cellular phenotypes and responses to perturbations. The potential applications of Cell Painting are vast, from identifying novel drug candidates to understanding disease mechanisms at the cellular level (Bray et al., 2016; Gustafsdottir et al., 2013; Ljosa et al., 2013). Unfortunately, fully realizing the potential of Cell Painting is hindered by significant challenges in data integration and interpretation. While recent approaches explore direct learning from cellular images, most established pipelines rely on extracted Cell Painting features (morphological measurements extracted from Cell Painting assays, in our case using CellProfiler software) Ando et al. (2017); Celik et al. (2022); Borowa et al. (2024); Korsunsky et al. (2019) - Typical Cell Painting datasets consist of thousands of such engineered features, leading to the "curse of dimensionality" and computationally intensive analyses (Caicedo et al., 2017). Moreover, these datasets are prone to batch effects - systematic variations unrelated to biology - due to differences in experimental conditions, imaging equipment, and data processing across laboratories (Singh et al., 2017).

To address these critical limitations, we propose CellPainTR, a novel foundational model designed to learn robust representations of Cell Painting features. By operating directly in the feature space, our approach uniquely integrates batch correction and representation learning into a single model. Through a three-stage training process, CellPainTR learns to distinguish batch-specific artifacts from biologically meaningful features, enabling adaptive balancing of batch effect correction and preservation of underlying biological signals. Leveraging a Transformer-like architecture (Vaswani et al., 2017) and Hyena operators (Poli et al., 2023), the model efficiently handles long-range dependencies

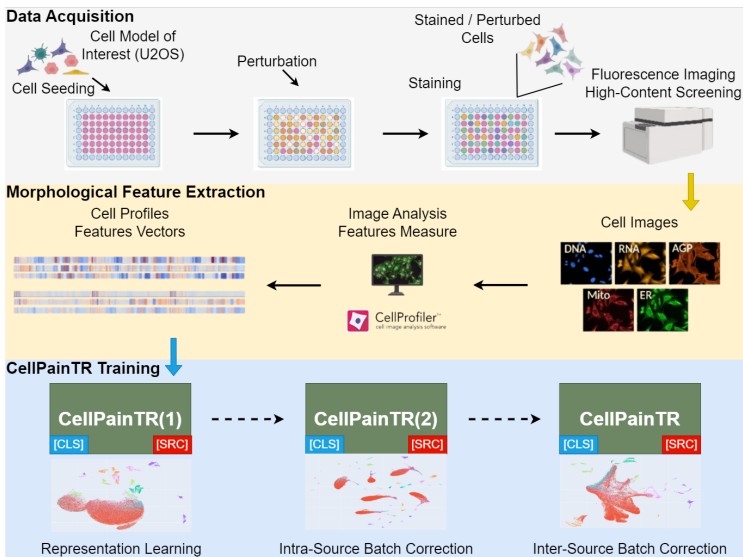

Figure 1: **Cell Painting workflow and CellPainTR training**. *Data acquisition* – (a) Cell Model Preparation: Cells of interest are seeded into multi-well plates. (b) Perturbation: Cells undergo various treatments or perturbations (e.g., drug compounds, genetic modifications). (c) Staining: Cells are stained with a mix of fluorescent dyes to highlight different cellular components. (d) Imaging: High-content screening using fluorescence microscopy captures detailed images of the stained cells. *Morphological feature extraction* – (e) Image Analysis: Software like CellProfiler processes the raw images to extract quantitative features, resulting in high-dimensional feature vectors for each cell. *CellPainTR training* – (f) CellPainTR Processing: The extracted feature vectors are input into CellPainTR, our novel Transformer-based model designed for unified batch correction and representation learning. Finally, CellPainTR produces batch-corrected and dimensionality-reduced representations of the cell profiles, enabling more robust integration and analysis of multi-source Cell Painting data.

and high-dimensionality inherent in Cell Painting data while maintaining direct compatibility with established analysis workflows and preserving the interpretability critical to biological research.

We evaluate the effectiveness of CellPainTR using a comprehensive set of metrics, including batch correction indices and compound-specific pattern preservation metrics from the Broad Institute benchmark (Arevalo et al., 2024). Experimental result 1 demonstrate that CellPainTR significantly outperforms existing approaches such as Combat and Harmony (Johnson et al., 2007; Korsunsky et al., 2019) across multiple evaluation metrics. Key results include superior batch correction scores (0.80 with controls, 0.69 without controls) compared to Combat (0.56, 0.37) and Harmony (0.57, 0.40), while maintaining strong biological information retention as evidenced by improved clustering metrics and qualitative UMAP visualizations. The versatility of our foundational model allows it to be fine-tuned for specific tasks and adapted to diverse experimental setups, enhancing its utility across different applications.

This work has significant implications for the broader field of image-based profiling and machine learning in biological research. By providing a more robust and adaptable solution to the batch effect problem, CellPainTR paves the way for more reliable and scalable analyses of Cell Painting data, with the potential to accelerate drug discovery and deepen our understanding of cellular biology.

## 2 BACKGROUND

**Classical Batch Correction Methods.** The challenge of batch effects in high-dimensional biological data, particularly in the field of image-based profiling like Cell Painting, has been a significant focus of research in recent years (Arevalo et al., 2024; Ando et al., 2017; Celik et al., 2022; Kraus

et al., 2024; Borowa et al., 2024). Several classical algorithms have been developed to address batch effects in biological data. One prominent example is Combat, which uses a parametric empirical Bayes framework to adjust for batch effects (Johnson et al., 2007). The Combat method models the batch effect as an additive and multiplicative effect on the data, and estimates these effects using an empirical Bayes approach. This can be expressed mathematically as:

$$y_{ij} = \alpha_i + \beta_i x_j + \gamma_b + \delta_b x_j + \epsilon_{ij} \tag{1}$$

where $y_{ij}$ is the observed data, $x_j$ is the covariate of interest, $\alpha_i$ and $\beta_i$ are the sample-specific intercept and slope, $\gamma_b$ and $\delta_b$ are the batch-specific intercept and slope, and $\epsilon_{ij}$ is the residual error.

Another method, Harmony, uses iterative clustering and linear adjustment to correct batch effects in multi-modal single-cell data (Korsunsky et al., 2019). Harmony aims to identify a shared low-dimensional representation across batches by aligning cluster centroids in an iterative fashion. This can be expressed as:

$$z_i = W_b h_i + b_b \tag{2}$$

where $z_i$ is the corrected representation for sample $i$, $h_i$ is the original high-dimensional representation, $W_b$ and $b_b$ are the batch-specific linear transformation parameters.

While these classical methods have demonstrated effectiveness in many scenarios, they may struggle to handle the high dimensionality and complex batch effects present in Cell Painting datasets (Singh et al., 2024).

**Self-Attention and the Hyena Operator.** The self-attention operator (Vaswani et al., 2017) is a fundamental mechanism of Transformers. Specifically, given a sequence $x \in \mathbb{R}^{L \times D}$ with length $L$ and $D$ features, the self-attention operator $A(x)$ is defined as:

$$A(x) = \sigma(xW_q)(xW_k)^T(xW_v) \tag{3}$$

where $W_q, W_k, W_v \in \mathbb{R}^{D \times D}$ are learnable projection matrices, and $\sigma$ is a softmax operation. This allows the model to capture pairwise relationships between tokens in the sequence. However, one limitation is that self-attention becomes computationally expensive for long sequences, with a complexity of $\mathcal{O}(L^2)$. To address the computational challenge of self-attention, the Hyena operator (Poli et al., 2023) was introduced as a replacement for self-attention in Transformers. The Hyena operator is characterized by a structured self-attention mechanism that involves long convolutions and element-wise gating:

$$y_t = (h * u)_t = \sum_{\tau=0}^{L-1} h_{t-\tau} u_\tau. \tag{4}$$

In standard convolutional architectures, the filter length $\ell$ is typically constrained by $\ell \ll L$, where $L$ is the input sequence length. This constraint helps control computational costs. However, by parameterizing the filter as a function of the temporal offset $\tau$ (i.e., $h_\tau = \gamma_\theta(\tau)$), we can design extended convolution kernels without a proportional increase in parameters. This technique, known as implicit convolution, enables efficient modeling of long-range dependencies. The implicit convolution mechanism is exemplified by the Hyena operator, which employs a recursive framework incorporating extended convolutions and point-wise modulation:

$$y = x^N \cdot (h^N * (x^{N-1} \cdot (h^{N-1} * (\cdots x^1 \cdot (h^1 * v))))) \tag{5}$$

Here, $v$ denotes the initial input, $\{x^i\}_{i=1}^N$ represent successive transformations, $N$ indicates the recursion depth, $*$ symbolizes convolution, and $\cdot$ denotes Hadamard (element-wise) product.

**Relevance to CellPainTR.** The Hyena operator's ability to efficiently model long-range dependencies in high-dimensional data makes it particularly relevant to the CellPainTR model. By incorporating the Hyena operator, CellPainTR can effectively handle the complex feature interactions present in Cell Painting data, which is critical for addressing batch effects while preserving biologically relevant information. Furthermore, the Bidirectional Hyena (Oh et al., 2023) extension, which removes the temporal causality, is particularly well-suited for the Cell Painting domain, where morphological feature interactions are not constrained by sequential order. This non-causal, bidirectional mechanism aligns with the requirements of the CellPainTR model.

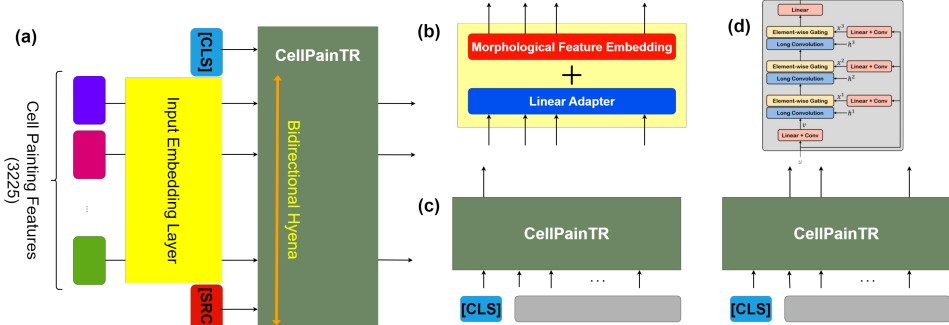

Figure 2: **CellPainTR architecture**. (a) CellPainTR incorporates several novel innovations: a linear adaptor layer combined with Morphological Feature Embeddings, a source context token (SRC), and a bidirectional Hyena operator. (b) Input embedding layer architecture. (c) After multi-step training, CellPainTR can be applied to various downstream tasks, including compound classification, compound retrieval, qualitative analysis, and batch correction. (d) Hyena Operator architecture.

## 3 CELLPAINTR

This section describes our novel feature embedding approach to unified batch correction and representation learning for Cell Painting data. We introduce a Transformer-like architecture that effectively handles the high dimensionality and complex relationships inherent in Cell Painting features while simultaneously addressing batch effects and preserving compound-specific molecular mechanisms of action (MoA) patterns (Arevalo et al., 2024; Moshkov et al., 2022; Chen et al., 2023).

CellPainTR's architecture and training process are specifically designed to address the unique challenges presented by Cell Painting data (see Fig. 2(a)). The high dimensionality of the data is tackled through the use of Hyena operators, which efficiently capture long-range dependencies across the extensive feature space. The linear adaptor for morphological feature embedding ensures that the full spectrum of continuous features is preserved, addressing the challenge of information loss often associated with discretization methods. The feature context embedding mechanism replaces traditional positional encoding, better capturing the intrinsic relationships between morphological features that are critical in Cell Painting data. Finally, the source-specific token and multi-stage training process directly address the batch effect problem by learning to distinguish between source-specific variations and true biological signals. This comprehensive approach enables CellPainTR to simultaneously correct for batch effects and learn biologically meaningful representations, a key requirement for effective analysis of multi-source Cell Painting datasets. More details are as follows.

### 3.1 DESIGN ARCHITECTURE

**Linear Adapter for Morphological Feature Embedding.** Cell Painting data consists of thousands of morphological features, each representing a specific aspect of cellular structure and organization. To effectively capture the rich information in this high-dimensional data, we introduce a linear adaptor module to embed the continuous features without any loss of information (Weisbart et al., 2024). Unlike traditional approaches that rely on discrete token representations, our linear adaptor maps the original feature space directly to the model's input embeddings as shown in Figure 2(b). This allows the CellPainTR model to operate on the full spectrum of the morphological features, preserving the nuanced relationships between them. For each feature $i$, the embedding is computed as:

$$E_i = \mathbf{W}_i \cdot C_i + \mathbf{b}_i \quad \text{where } E_i \in \mathbb{R}^{d_{\text{model}}} \tag{6}$$

where $\mathbf{W}_i \in \mathbb{R}^{d_{\text{model}}}$ is a learnable weight matrix, $C_i \in \mathbb{R}$ is the input feature value, and $\mathbf{b}_i \in \mathbb{R}^{d_{\text{model}}}$ is a learnable bias vector. This linear adaptor approach ensures that the CellPainTR model can effectively leverage the full contextual information present in the Cell Painting data, laying the foundation for robust batch correction and representation learning (Tromans-Coia & Jamali, 2023).

**Feature Context Embedding.** Traditional Transformer models use positional encoding to incorporate the sequential nature of the input data. However, in the case of Cell Painting features, the order

of the features does not necessarily reflect any meaningful biological context. To better capture the intrinsic relationships between the morphological features, we introduce a feature context embedding mechanism (Seal et al., 2024), as shown in Figure 2(b). Specifically, the feature context embedding replaces the standard positional encoding by learning an embedding of the feature context. In this approach, each morphological feature is encoded with its own embedding $(M_1, M_2, ..., M_L)$ with $M_i \in \mathbb{R}^{1 \times d_{\text{model}}}$, which is then added to the expression embeddings. The complete feature embedding matrix is then constructed by concatenating all feature embeddings:

$$\mathbf{E} = [M_1; M_2; \ldots; M_L] \in \mathbb{R}^{L \times d_{\text{model}}} \tag{7}$$

where $L$ is the number of morphological features and $d_{\text{model}}$ is the embedding dimension, and $[\cdot; \cdot]$ denotes concatenation along the row direction. This method allows us to provide the CellPainTR model with explicit feature context information. By using the feature context embedding, the CellPainTR model can learn to associate the morphological features with their biological context, rather than relying on their position in the input sequence. This enhances the model's ability to capture the complex interdependencies between the features, which is crucial for effective batch correction and representation learning.

**Source Context Token.** Cell Painting datasets often originate from multiple experimental sources, each with its own unique batch-related characteristics. To explicitly model these source-specific variations, CellPainTR incorporates a special source context token (Weisbart et al., 2024). The source context token $S$ with dimension $S_{dim} = M_{dim}$ is initialized as a learnable parameter, drawn from its own embedding with a vocabulary size $K$ with $K$ the number of source in the dataset; $(S_1, S_2, ..., S_K)$ and is concatenated with the input feature embeddings, as shown in Figure 2(a). We incorporate a learnable source embedding that is concatenated to the feature embeddings:

$$\mathbf{S}_k = \text{Embedding}(k) \quad \text{where } k \in \{1, \ldots, K\}, \mathbf{H} = [\mathbf{E}; \mathbf{S}_k] \quad \text{where } \mathbf{S}_k \in \mathbb{R}^{1 \times d_{\text{model}}} \tag{8}$$

where $K$ is the number of sources in the dataset. During training, the model learns to associate the source context token with the unique batch effects present in each data source. This allows the model to adaptively correct for batch-related biases while preserving the biologically relevant information in the learned representations. The inclusion of the source context token is a key innovation that enables the CellPainTR model to handle the challenges of integrating Cell Painting data from diverse experimental sources, a critical requirement for advancing drug discovery and cellular biology research (Tromans-Coia & Jamali, 2023).

## 3.2 TRAINING

The CellPainTR model is trained using a multi-stage process that combines unsupervised and supervised learning objectives to achieve unified batch correction and representation learning. By progressively exposing the model to increasingly diverse data contexts, we enable it to learn robust, generalizable representations that preserve compound-specific MoA relationships while mitigating batch-related confounders. More details are as follows.

**Channel-Wise Masked Morphology (CWMM).** The initial stage of training utilizes Channel-Wise Masked Morphology (CWMM), a novel approach inspired by Masked Language Modeling (MLM) in natural language processing and Masked Expression Modeling (MEM) in single-cell RNA sequencing analysis (Oh et al., 2023). CWMM is tailored to handle the continuous values of morphological features in Cell Painting data while respecting its channel-wise structure (Seal et al., 2024). For additional details on CWMM refer to Appendix A.1

As shown in Fig. 3(a), in the CWMM task, a subset of input morphological features is randomly masked, with the model tasked to predict these masked values based on the surrounding context. Features are grouped based on both their channel origin and the cellular compartment they describe (e.g., "DNA channel - Nucleus" features form one group, "Mito channel - Cytoplasm" features form another). The masking probability for each training batch is chosen from a range of $[0.05, 0.4]$ and is applied uniformly across all feature sets, preserving the biological relationships within each channel-compartment combination. Importantly, only non-zero values are masked and replaced with a [MASK] token, as distinguishing between true and false zero values is not feasible in this context (Way et al., 2021).

Figure 3: **CellPainTR training**. (a) Channel-Wise Masked Morphology (CWMM) for self-supervised learning. (b) Intra-Source Supervised Contrastive Learning: combining CWMM and single source supervised contrastive learning. (c) Inter-Source Supervised Contrastive Learning: combining CWMM and multi source supervised contrastive learning.

Mathematically, the objective function for the CWMM pre-training task is formulated as:

$$\ell_{\text{CWMM}} = \frac{1}{G} \sum_{g=1}^{G} \frac{1}{|M_g|} \sum_{i \in M_g} (F_{g,i} - F'_{g,i})^2 \tag{9}$$

With $G$ the number of feature groups (channel-compartment combinations), $M_g$ represents the set of masked indices for feature group $g$, $|M_g|$ the number of masked features in group $g$, $F_{g,i}$ denotes the true value of the $i$-th morphological feature in group $g$, $F'_{g,i}$ the predicted value for the $i$-th masked feature in group $g$. Through this pre-training process, CellPainTR acquires generalizable features that capture the biological meaning encoded in the morphological data, the contextual relationships between features, and the channel-wise dependencies within the Cell Painting data structure. This comprehensive understanding of the intricate patterns and correlations present in Cell Painting data establishes a robust foundation for the subsequent supervised learning stages, ultimately enhancing the model's capacity for batch correction and representation learning. For additional details, please refer to Appendix A.2

**Intra Source Supervised Learning.** Following the unsupervised CWMM pretraining, the model undergoes a fine-tuning phase using a supervised contrastive learning approach within each data source. This intra-source supervised learning stage is designed to encourage the model to learn representations that are both discriminative and invariant to batch-related variations within a given experimental source (plate effects, liquid handling, reagent batches, ...). As shown in Fig. 3(b), during this phase, we allow biological feature metadata to flow to the model during training (more specifically InChIKey via the supervised contrastive objective), while ensuring that each batch contains data from only a single source. The objective function for this stage combines the CWMM loss with a supervised contrastive loss, weighted equally:

$$\ell_{\text{intra}} = \ell_{\text{CWMM}} + \ell_{\text{supcon}} \tag{10}$$

The supervised contrastive loss $\ell_{\text{supcon}}$ is computed directly using the CLS token output of the encoder. This can be expressed as:

$$\ell_{\text{supcon}} = -\sum_{i \in I} \frac{1}{|P(i)|} \sum_{p \in P(i)} \log \frac{\exp(h_i \cdot h_p / \tau)}{\sum_{a \in A(i)} \exp(h_i \cdot h_a / \tau)} \tag{11}$$

Here, $I$ is the set of indices, $P(i)$ is the set of positives for sample $i$ (i.e., samples with the same biological label - InChIKey), $A(i)$ is the set of all samples except $i$, $h_i$ and $h_p$ are the normalized CLS token outputs for samples $i$ and $p$, respectively, and $\tau$ is a temperature parameter. This approach allows the model to learn representations that capture biologically relevant information while remaining robust to source-specific batch effects. For additional details, please refer to Appendix A.3

**Inter Source Supervised Learning.** The final stage of training involves fine-tuning the model using a supervised contrastive learning objective that spans multiple data sources. This inter-source supervised learning step enables the model to learn representations that are not only batch-corrected but also generalize well across diverse experimental conditions and data sources.

As shown in Fig. 3(c), the key distinction in this stage is that we now allow different sources to be mixed within the same batch, while source context token specific to each source are optimized during the training. Thanks to the source context token, the supervised contrastive loss now operates across samples from multiple sources, encouraging the model to learn representations that are invariant to source-specific variations (microscope settings, experimental setups, ...) while still capturing biologically meaningful information. For additional details, please refer to Appendix A.4

## 4 EXPERIMENTS

Our experimental framework was designed to rigorously evaluate the effectiveness of our proposed method for batch correction and representation learning in Cell Painting data. We conducted a series of experiments using a large-scale dataset, applying a multi-stage training process and assessing performance through both qualitative and quantitative measures. Qualitatively, we examined the model's ability to preserve clustering patterns that align with known compound Mechanisms of Action (MoA). Quantitatively, we used established metrics from the Broad Institute benchmark (Arevalo et al., 2024), to measure the preservation of compound-specific cellular responses.

**Dataset.** For our experiments, we utilized the cpg-0016 (Chandrasekaran et al., 2023) Cell Painting dataset from the JUMP consortium. This extensive dataset encompasses a diverse array of biological perturbations, including over 100,000 small molecule compounds and genetic perturbations such as ORF overexpression and CRISPR knockouts targeting thousands of genes. The dataset captures a wide spectrum of morphological features, typically numbering in the thousands per cell, which provide detailed measurements of cell shape, size, texture, and intensity across multiple cellular compartments (Bray et al., 2016). The dataset includes both negative and positive controls, which are crucial for establishing baselines and validating assay performance. Negative controls, such as non-targeting controls or DMSO treatments, provide a reference point for normal cellular morphology (Caie et al., 2010). Positive controls, consisting of known bioactive compounds or genetic perturbations with well-characterized effects, serve to validate the assay's sensitivity and specificity (Gustafsdottir et al., 2013).

Our preprocessing pipeline was designed to address common challenges in Cell Painting data. First, we imputed all infinite and missing values with zero to ensure computational feasibility. Next, we applied MAD (Median Absolute Deviation) normalization, using the negative control of each plate as the baseline. This step helps to mitigate plate-to-plate variations and standardize feature scales. Finally, we employed a clipping strategy, constraining values between the 0.01 and 0.99 quantiles. This step was crucial in managing extreme outliers, which are common artifacts in Cell Painting data and can significantly impact model performance if left unaddressed.

To evaluate the robustness and generalizability of our model, we experimented with various train-test split strategies. These included hiding certain data source generations from training, excluding specific plates, and random partitioning of the data (Goodfellow et al., 2016). This approach allowed us to choose the best-performing model under different scenarios of data availability and batch effects.

**Trainining.** The pretraining phase of our model utilized all compound-related data from the dataset. During this phase, the model learned to predict masked morphological features based on the surrounding context, leveraging the bidirectional nature of the Hyena blocks. This approach enabled the model to capture complex relationships within the data effectively, laying the groundwork for subsequent supervised learning stages (Devlin et al., 2018).

Following pretraining, we moved to a more focused training phase using a curated subset of the data. This subset concentrated on compounds with rich metadata, including control compounds and those with known Mechanisms of Action (MoA) (Schurer et al., 2011). This phase, which took approximately one week on the same hardware, allowed the model to refine its representations based on more specific biological contexts (Goodfellow et al., 2016). During this stage, we introduced the source-specific token, enabling the model to learn and account for source-specific variations in the data (Devlin et al., 2018). This approach was crucial in addressing batch effects while preserving biologically relevant information.

The final stage of our training process involved fine-tuning the model with a strong emphasis on contrastive learning. This phase also lasted approximately one week and used the same curated

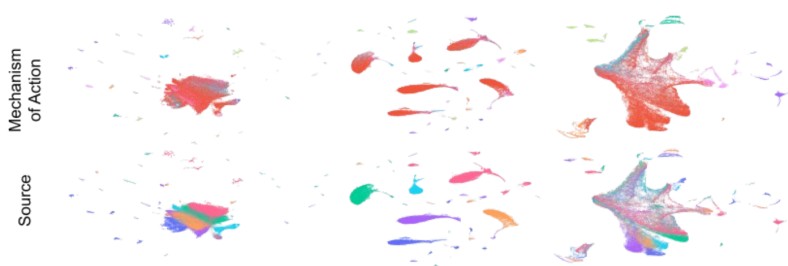

Figure 4: **UMAP visualizations**. Comparing Baseline (uncorrected), CellPainTR at its second training step, and the final CellPainTR model. Columns represent methods or training stages, and rows depict data aspects. The top row uses MoA coloring for phenotypic variations, with red for negative controls. The bottom row uses source coloring for batch effects. The Baseline (left) shows strong batch effects and weak spatial constraints in compound clusters, indicating a lack of robust biological signals. CellPainTR's improvement is seen in the middle and right columns. The second step (middle) shows clear source-based clusters with reduced intra-source fragmentation. The final step (right) achieves cohesive batch integration and clear compound-specific patterns, supported by superior Batch Correction (0.76) and Graph Connectivity (0.84) scores, demonstrating effective batch variability mitigation while preserving biological signals.

dataset as the training phase (Schurer et al., 2011). The contrastive learning approach encouraged the model to learn representations that are invariant to batch effects while still capturing meaningful biological differences (Chen et al., 2020). By contrasting samples from different sources but with similar biological properties, the model learned to distinguish between batch-related variations and true biological signals (He et al., 2020).

### 4.1 QUALITATIVE EVALUATION RESULTS

Figure 4 illustrates CellPainTR's effectiveness in batch correction and biological signal preservation for Cell Painting data. The visualizations demonstrate the model's ability to account for batch variability stemming from different laboratories, batches and microscopes while maintaining crucial biological information. The top row uses compound coloring to represent biological variation, with red indicating the negative control. The bottom row employs source coloring to highlight batch effects. In the compound-colored row (top), CellPainTR demonstrates progressively clearer separation of compounds (each represented by a unique color), especially in the final stage (right). This improved clustering reflects an enhanced preservation of biological information. Simultaneously, the source-colored row (bottom) highlights CellPainTR's effectiveness in mitigating source-specific batch effects, with the final stage showing the most integrated distribution of data points across sources. The improved clustering and reduced overlap in CellPainTR's final stage, compared to both the baseline and its intermediate stage, underscore its superior performance in balancing batch correction with biological signal retention. This visualization demonstrates CellPainTR's superior performance in balancing batch correction with biological signal preservation, addressing a key challenge in integrating high-dimensional microscopy data from diverse sources. For more detailed comparison, please refer to Appendix E

### 4.2 QUANTITATIVE EVALUATION RESULTS

Table 1 presents a comprehensive comparison of batch correction methods, including our proposed CellPainTR approach, across various metrics. These metrics can be broadly categorized into: (1) batch correction measures, which assess the removal of technical variations; (2) compound-specific pattern preservation indicators (labeled as 'Biological Metrics' in the table), which evaluate how well the method maintains compound-related effects and known mechanism of action relationships; and (3) aggregate scores, which average each and both aspects. For consistent interpretation, all metrics have been normalized to a scale of 0 to 1, where 0 indicates poor performance and 1 represents optimal performance C.2.

Table 1: Performance comparison of batch correction methods

| Method | Batch Correction | | | | Biological Metrics | | | | | | | Aggregate Scores | | |
|---|---|---|---|---|---|---|---|---|---|---|---|---|---|---|
| | Graph Conn. | Sil. Batch | Batch Corr. (control) | Batch Corr. (no control) | Leiden NMI | Leiden ARI | Sil. Label | mAP (control) | mAP (no rep) | Bio Info (control) | Bio Info (no control) | Batch Corr. | Bio Metrics | Overall Score |
| Baseline | **0.86** | 0.93 | 0.57 | 0.40 | 0.41 | 0.20 | 0.50 | 0.48 | 0.58 | **0.91** | 0.86 | 0.69 | 0.56 | 0.63 |
| Combat Johnson et al. (2007) | 0.85 | 0.93 | 0.56 | 0.37 | 0.39 | 0.12 | 0.50 | 0.48 | 0.58 | **0.91** | 0.85 | 0.68 | 0.55 | 0.61 |
| Harmony Korsunsky et al. (2019) | 0.80 | 0.93 | 0.57 | 0.40 | 0.42 | 0.24 | 0.50 | 0.47 | 0.58 | **0.91** | 0.86 | 0.68 | 0.57 | 0.62 |
| Sphering Kessy et al. (2018) | 0.64 | **0.95** | 0.70 | 0.58 | 0.36 | 0.35 | 0.48 | 0.12 | 0.23 | 0.66 | 0.85 | 0.72 | 0.43 | 0.58 |
| CellPainTR(1) | 0.78 | 0.73 | 0.78 | **0.69** | 0.34 | **0.26** | 0.52 | 0.22 | 0.32 | 0.72 | 0.81 | 0.75 | 0.46 | 0.60 |
| CellPainTR(2) | 0.69 | 0.75 | 0.71 | 0.58 | **0.43** | 0.15 | **0.70** | **0.54** | **0.63** | 0.84 | **0.91** | 0.68 | **0.60** | **0.64** |
| CellPainTR | 0.84 | 0.70 | **0.80** | **0.69** | 0.35 | 0.17 | 0.57 | 0.40 | 0.54 | 0.86 | 0.79 | **0.76** | 0.53 | **0.64** |

To assess the effectiveness of batch correction, we employed several complementary metrics. Graph Connectivity evaluates the preservation of biological relationships across batches by measuring the proportion of k-nearest neighbors that are maintained after batch correction, compared to the original data structure. This metric provides insights into how well the local structure of the data is preserved while removing batch effects, with values closer to 1 indicating better preservation of biological relationships. The Silhouette Batch score quantifies the degree of separation between batches by comparing the mean distance between samples from different batches to the mean distance within batches, where values closer to 1 suggest more effective batch integration. Additionally, we introduce a Batch Correction metric calculated as 1 - f1 score, where f1 represents the performance of a classifier trained to detect batch effects in the corrected data. In this context, higher values indicate superior batch correction as the classifier struggles to distinguish between batches, effectively measuring the degree of batch effect removal.

The preservation of compound-specific patterns is crucial in ensuring that the batch correction process does not compromise the underlying MoA-related information. We quantify this preservation through multiple complementary approaches. Through the Leiden clustering algorithm, we compute Normalized Mutual Information (NMI) and Adjusted Rand Index (ARI) scores by comparing cluster assignments with known biological labels. These metrics provide different perspectives on biological preservation: NMI values range from 0 to 1, indicating the degree of shared information between cluster assignments and biological labels, while ARI provides a chance-corrected measure of agreement, with values above 0.3 indicating strong biological preservation. The Silhouette Label score further strengthens our evaluation by assessing how well samples cluster according to biological conditions rather than batch effects, with values above 0.5 suggesting robust biological signal retention. To evaluate the impact on downstream analysis tasks, we employed mean Average Precision (mAP) scores (see Appendix C.1)for compound retrieval performance in two contexts: with controls (mAP control) and without replicate compounds (mAP no rep). These scores specifically assess the model's ability to identify similar biological conditions across batches, providing a practical measure of biological signal preservation where values above 0.5 indicate strong retention of compound-specific patterns. Aggregate scores provide a holistic view of each method's performance. The Batch Correction score summarizes the overall effectiveness in removing batch effects, while the Biological Metrics score encapsulates the method's ability to preserve biological information. The Overall Score combines these aspects to give a comprehensive assessment of each method's performance.

Our proposed CellPainTR method demonstrates strong performance across these metrics. Notably, it achieves the highest batch correction scores both with and without controls, indicating superior batch integration capabilities. The progressive improvement observed across the three steps of CellPainTR (denoted as CellPainTR(1), CellPainTR(2), and CellPainTR) highlights the method's ability to balance batch effect removal with biological information preservation. Specifically, CellPainTR consistently outperforms traditional methods like Combat and Harmony in key metrics. While Combat and Harmony show strengths in certain areas, such as graph connectivity and silhouette batch scores, CellPainTR demonstrates a more balanced performance across all metrics. This is particularly evident in the higher batch correction scores and competitive biological metric scores. The final CellPainTR model achieves the highest overall score, tied with CellPainTR(2), suggesting that our approach successfully addresses batch effects while maintaining crucial biological information. This balance is critical in ensuring that batch correction does not come at the cost of losing important biological signals. In summary, the quantitative results presented in Table 1 provide strong evidence for the efficacy of CellPainTR in batch correction tasks. The method's ability to consistently perform well across various metrics, particularly in batch correction and biological information retention, po-

sitions it as a robust solution for addressing batch effects in Cell Painting data analysis. For further details on the metrics implementation please refer to Appendix C.

### 4.3 ABLATION STUDY

To elucidate the contribution of each component in our CellPainTR method, we conducted an ablation study by evaluating the model's performance at different stages of training. The results are presented in Table 1, where CellPainTR(1) and CellPainTR(2) represent intermediate stages, and CellPainTR denotes the final model.

The initial training stage, represented by CellPainTR(1), demonstrated a strong focus on batch effect removal. This stage achieved the highest batch correction score without control (0.69), surpassing both the baseline and traditional methods in this metric. However, this came at the cost of reduced performance in biological metrics, particularly evident in the lower mAP scores (0.22 and 0.32) and decreased graph connectivity (0.78) compared to the baseline (0.86). CellPainTR(2), the intermediate training stage, marked a significant shift towards preserving biological information. This stage showed substantial improvements in biological metrics, most notably in Silhouette Label (0.70) and mAP scores (0.54 and 0.63). Moreover, it achieved the highest Leiden NMI (0.43) among all methods, suggesting improved cluster separation. These gains in biological signal preservation were accompanied by the best overall biological metrics score (0.60). However, this stage also saw a decrease in batch correction performance compared to CellPainTR(1) and a further reduction in graph connectivity (0.69). The final CellPainTR model emerged as a balanced solution, optimizing both batch correction and biological signal preservation. It achieved the best overall batch correction score (0.76), effectively balancing correction with and without control. The model also recovered graph connectivity (0.84) compared to the intermediate stages while maintaining strong performance in biological information preservation (0.86 for control). Despite these improvements, there was a slight decrease in some biological metrics compared to CellPainTR(2), particularly in Silhouette Label and Leiden NMI.

This ablation study reveals a critical trade-off between batch correction and biological signal preservation throughout the training process. The initial stage prioritizes batch effect removal, potentially at the expense of biological signal retention. The intermediate stage then shifts focus to preserving biological information, enhancing clustering and representation quality. Finally, the complete model achieves an optimal balance between these competing objectives. For qualitative comparison of the steps, please refer to Appendix E

## 5 CONCLUSION

This paper introduces CellPainTR, a novel approach to unified batch correction and representation learning for high-dimensional Cell Painting data. Leveraging a Transformer-like architecture with Hyena operators, CellPainTR addresses the critical challenges of batch effects and dimensionality reduction in image-based profiling. Extensive experimental results confirmed that CellPainTR successfully addresses batch effects while preserving crucial biological information. However, we acknowledge the limitations of our study, including the restricted comparison to only three other batch correction methods due to difficulties in replicating results from benchmark papers (Arevalo et al., 2024). Additionally, we observed a discrepancy between the qualitative improvements seen in the UMAP visualizations and some of the quantitative metrics, highlighting the complexity of evaluating batch correction methods and the potential limitations of current evaluation approaches (for more details see Appendix D). Despite these challenges, CellPainTR represents a significant step forward in the analysis of Cell Painting data, offering a powerful approach to integrating multi-source datasets while maintaining compound-specific molecular signature preservation. The method's ability to handle high-dimensional data, correct for batch effects, and preserve biological information positions it as a valuable tool for advancing drug discovery and cellular biology research.

### ETHICS STATEMENT

The utilization of models such as CellPainTR offers significant benefits for advancing our understanding of complex biological systems and potentially improving medical research. However, eth-

ical considerations must guide its use to ensure responsible data handling, avoid biases, and protect individual privacy, underscoring the importance of ethical guidelines and regulations in the application of such models.

## REPRODUCIBILITY STATEMENT

We provide detailed implementation information in Section 3.2 and additional details in Appendix A. A comprehensive description of the datasets used in our experiments can be found in Section 4 and using the official dataset link: https://github.com/jump-cellpainting/datasets and precision about dataset curation can be found in Appendix B. Our source code is available for access at the following link: https://github.com/CellPainTR/CellPainTR.

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

## A  EXPERIMENTAL DETAILS

For all downstream tasks, we fine-tuned the model using the AdamW optimizer (Loshchilov & Hutter, 2019). The fine-tuning process was executed using one NVIDIA Quadro RTX 4000.

### A.1  CHANNEL-WISE MASKED MORPHOLOGY (CWMM)

Channel-Wise Masked Morphology (CWMM) is a novel training objective designed for Cell Painting data analysis. Inspired by Masked Language Modeling (MLM) (Devlin et al., 2018) in natural language processing and Masked Expression Modeling (MEM) in single-cell RNA sequencing (Oh et al., 2023), CWMM adapts these concepts to address the continuous values and channel-wise structure of morphological features. In CWMM, a subset of input morphological features is randomly and uniformaly masked with respect to each channel during training. The model then predicts these masked values based on surrounding context. Key aspects of the implementation include:

- Masking probability: Randomly selected from a range of 0.05 to 0.4 for each training batch.
- Uniform application: Masking is applied equally across all five channels in Cell Painting data.
- Non-zero value focus: Only non-zero values are masked and replaced with a [MASK] token.

The CellPainTR model, using bidirectional Hyena blocks, predicts the true values of masked features. This process requires consideration of both preceding and following features, capturing complex relationships within the data. The CWMM objective function is formulated as in Section 3.2 Eq. (9).

CWMM offers several advantages:

- Encourages learning of generalizable features capturing biological meaning in morphological data.
- Facilitates understanding of contextual relationships and channel-wise dependencies.
- Establishes a foundation for subsequent supervised learning, enhancing capabilities in batch correction and representation learning.

By adapting proven techniques to Cell Painting data's specific challenges, CWMM provides a powerful tool for extracting insights from complex morphological datasets. This approach has the potential to advance our understanding of cellular phenotypes and their relationships to underlying biological processes, benefiting fields from drug discovery to basic cell biology research.

### A.2  PRETRAINING - STEP 1

The pretraining phase of our model utilized all compound-related data from the dataset. We employed a configuration with 3 recurrences and 4 Hyena layers, training for 3 epochs. The model was optimized using AdamW (Devlin et al., 2018) with a learning rate of 1e-4 and a batch size of 16 for

approximately two weeks. The model also had access to the source context token, as described in Section 3.1, but had no constraint or direct objective regarding this supplementary context. During this step, all compound data was used, including negative controls, positive controls, and unknown compounds. This approach enabled the model to effectively capture complex relationships within the data, facilitating the learning of lower-dimensional representations.

### A.3 PRETRAINING - STEP 2

The second training phase focused on compounds with rich metadata, including controls and those with known Mechanisms of Action (MoA). We reduced the learning rate to 1e-5, increased the batch size to 32, and ran this phase for approximately one week. This allowed the model to refine its representations based on more specific biological contexts. During this stage, we introduced intra-source supervised contrastive learning alongside a reconstruction constraint (Channel-Wise Masked Morphology). To implement this, we restricted each batch to a single source, ensuring positive and negative pairs were intra-source. The first epoch served as a warm-up, allowing backpropagation only through the learnable source context token. Subsequently, we unfroze the rest of the model for the remaining epochs. This dual approach enabled the model to account for source-specific variations while preserving biological signals, effectively addressing intra-source batch effects without losing relevant information.

### A.4 PRETRAINING - STEP 3

The final stage of our training process involved fine-tuning the model with a strong emphasis on contrastive learning. We maintained the learning rate at 1e-5 but increased the batch size to 64. This phase, lasting approximately one week, used the same curated dataset as the second training phase. Unlike the previous step, we shuffled the data to include multiple sources in each batch, compelling the model to learn source-invariant representations. This approach aimed to preserve biological signals while mitigating inter-source batch effects.

## B DATASET

As supplementary context, drawn from the official dataset paper (Chandrasekaran et al., 2023) and the associated GitHub page, we provide additional details on dataset curation. As described in Appendix A, in the first step, we utilize all data related to compound perturbations. The full dataset includes Open Reading Frame (ORF) perturbations, Clustered Regularly Interspaced Short Palindromic Repeats (CRISPR) perturbations, and compound perturbations. Using the associated metadata files, we specifically select only the compound-related perturbations. In the second and third steps, we further curate the data by using the metadata to select compounds with a referenced Mechanism of Action (MoA), or those identified as positive controls or positive compound pairs.

## C BASELINE METHOD

All methods used in this study follow the implementation provided by the scib Python library , as described in the benchmark paper by (Arevalo et al., 2024).

### C.1 METRICS

Mean Average Precision (mAP) Evaluation:

1. Each sample in the dataset serves as a query

2. Positive elements: Other biological replicates sharing the same compound (identified by InChIKey)

3. Negative elements for mAP (control): Control wells from the same plate

4. Negative elements for mAP (no rep): Wells from the same plate treated with different compounds

5. Higher scores indicate better preservation of compound-specific biological effects after batch

The mean average precision (mAP) metric is also implemented in accordance with the benchmark paper (Arevalo et al., 2024).

For each query, there are $M - 1$ positive samples, which share the same compound, and $N$ negative samples, consisting of profiles from the same plate with different compounds or negative controls. A ranked list is generated using cosine similarity between the query profile and the $(M - 1) + N$ profiles. The average precision (AP) of the $i^{th}$ query is defined as:

$$AP_i = \sum_{k=1}^{(M-1)+N} (R_k - R_{k-1})P_k \tag{12}$$

where:

$$R_k = \frac{TP_k}{M - 1} \text{ is recall at rank k}$$

$$P_k = \frac{TP_k}{k} \text{ is precision at rank k}$$

$$TP_k \text{ is the number of positive elements retrieved up to rank k}$$

Finally, the mean average precision (mAP) for a compound is the average of the AP values across all replicates:

$$mAP = \frac{1}{M} \sum_{i=1}^{M} AP_i \tag{13}$$

### C.2    EVALUATION METRICS: MATHEMATICAL FORMULATIONS AND LIMITATIONS

#### C.2.1    ROLE OF CONTROLS IN METRIC CALCULATION

In our evaluation framework, metrics are reported in two variants: "with controls" and "without controls". This distinction relates directly to the negative controls present in each plate, which are also used as baseline in the Median Absolute Deviation normalization process.

**With Controls**    Metrics calculated "with controls" include negative control wells in the evaluation. Since these controls serve as the normalization baseline, they represent the most standardized condition across plates. Performance metrics in this setting typically show better results as these samples are inherently more aligned across batches due to their role in the normalization process.

**Without Controls**    The "without controls" variant excludes negative control wells, focusing exclusively on compound-treated samples. This provides a more stringent and realistic assessment of batch correction performance, as it evaluates the method's effectiveness on samples that weren't used in the normalization process.

**Interpretation Considerations**

1. Better performance in "with controls" metrics is expected due to the standardization process

2. "Without controls" metrics provide a more challenging benchmark for real-world application

3. The gap between these variants can indicate how well the batch correction generalizes beyond standardized samples

4. Methods showing consistent performance across both variants suggest robust batch correction capabilities

### C.2.2 BATCH CORRECTION METRICS

**Graph Connectivity**    Mathematical formulation:

$$GC = \frac{|\{(i,j) \in kNN_{before} \cap kNN_{after}\}|}{k \times n} \tag{14}$$

where $kNN_{before}$ and $kNN_{after}$ represent the k-nearest neighbors before and after batch correction, n is the number of samples, and k is the number of neighbors (we use k=15).

Limitations:

1. Sensitive to the choice of k
2. May not capture global structure changes
3. Can be biased by density differences between batches

**Batch Correction and Biological Information Scores**    For both metrics, we use a classifier but with different objectives and interpretations:

$$BC = 1 - F1_{batch\_classifier} \tag{15}$$
$$BI = F1_{compound\_classifier} \tag{16}$$

where $F1_{batch\_classifier}$ is the F1 score of the classifier trained to predict batch identifier and $F1_{compound\_classifier}$ is the F1 score for predicting InChIKeys (corresponding to different MoAs).

For a given classification task, the F1 score is calculated as:

$$F1 = 2 \times \frac{precision \times recall}{precision + recall} \tag{17}$$

The rationale behind these metrics is:

1. For Batch Correction (BC): Higher values (closer to 1) indicate better batch correction as the classifier fails to distinguish between batches
2. For Biological Information (BI): Higher values (closer to 1) indicate better preservation of biological signals as the classifier successfully distinguishes between compounds with different MoAs

Limitations:

1. May not capture very subtle batch effects or biological signals
2. Requires sufficient samples per batch/compound for reliable estimation
3. F1 score can be affected by the distribution of classes in the dataset

Alternative metrics: While established metrics like kBET exist in the field, technical limitations in available implementations at the time of development led to our current approach. Our metrics provide direct insight into both batch effect removal and biological signal preservation using the same classification framework.

### C.2.3 BIOLOGICAL METRICS

**Leiden Clustering Metrics (NMI & ARI)**    For two clusterings U and V:

$$NMI(U,V) = \frac{2 \times I(U,V)}{H(U) + H(V)} \tag{18}$$

where I(U,V) is the mutual information and H(U), H(V) are the entropies.

$$ARI = \frac{\text{RI} - E[\text{RI}]}{\max(\text{RI}) - E[\text{RI}]} \tag{19}$$

where RI is the Rand Index and E[RI] is its expected value.

Limitations:

1. Both metrics are sensitive to the number of clusters
2. ARI can be pessimistic with many small clusters
3. Results depend on Leiden algorithm parameters

**Mean Average Precision (mAP)**    For a query q:

$$AP(q) = \sum_{k=1}^{n} P(k) \times rel(k) \tag{20}$$

where P(k) is precision at cutoff k, rel(k) is relevance of k-th item.

Limitations:

1. Sensitive to the number of relevant items
2. May not capture complex biological relationships
3. Can be biased by compound representation imbalance

### C.2.4   EDGE CASES AND COMMON FAILURE MODES

1. Graph Connectivity may fail when:
   (a) Batch effects are confounded with biological signals
   (b) Data has very different scales across batches
   (c) Extreme outliers are present
2. Biological metrics may be unreliable when:
   (a) Very few replicates are available
   (b) Compound effects are subtle
   (c) Multiple mechanisms of action overlap
3. General considerations:
   (a) All metrics assume sufficient sample size per batch
   (b) Metrics may be less reliable with highly imbalanced batches
   (c) Strong batch effects might mask biological signals in all metrics

### C.2.5   AGGREGATED SCORES CALCULATION

The aggregated scores presented in Table 1 are calculated as weighted averages of their respective constituent metrics:

**Batch Correction Score**    The Batch Correction aggregate score is calculated as the arithmetic mean of the batch-related metrics:

$$BC_{score} = \frac{GC + S_{batch} + BC_{control} + BC_{no\_control}}{4} \tag{21}$$

where $GC$ is the Graph Connectivity, $S_{batch}$ is the Silhouette Batch score, $BC_{control}$ is the Batch Correction score with controls, and $BC_{no\_control}$ is the Batch Correction score without controls.

**Biological Metrics Score**    The Biological Metrics aggregate score combines all biology-related metrics:

$$Bio_{score} = \frac{NMI + ARI + S_{label} + mAP_{control} + mAP_{no\_rep} + BI_{control} + BI_{no\_control}}{7} \tag{22}$$

where $NMI$ is the Normalized Mutual Information from Leiden clustering, $ARI$ is the Adjusted Rand Index from Leiden clustering, $S_{label}$ is the Silhouette Label score, $mAP_{control}$ and $mAP_{no\_rep}$ are the Mean Average Precision scores with controls and without replicates respectively, and $BI_{control}$ and $BI_{no\_control}$ are the Biological Information scores with and without controls.

**Overall Score** The Overall Score is calculated as the arithmetic mean of the Batch Correction and Biological Metrics scores:

$$Overall_{score} = \frac{BC_{score} + Bio_{score}}{2} \tag{23}$$

This balanced approach ensures that both batch correction effectiveness and biological signal preservation contribute equally to the final evaluation of method performance.

### C.3 DATA PREPROCESSING

For the baseline data preprocessing shown in Table 1 (Baseline), we follow the steps outlined in the benchmark paper (Arevalo et al., 2024):

1. **Variation Filtering**: Features with low variance are filtered using the absolute coefficient of variation, with a threshold of $C_{\text{var}} < 1e^{-3}$.

2. **Median Absolute Deviation**: For each well, feature values are normalized using the median $\bar{X}$ and the absolute deviation $\tilde{\sigma}$, both calculated from control wells.

3. **Rank-based Inverse Normal Transformation (INT)**: Feature values are transformed based on their rank within each plate, following Blom's formula:

$$Y_i = \Phi^{-1}\left(\frac{r_i - c}{N - 2c + 1}\right)$$

   where $r_i$ is the rank of sample $i$, $N$ is the number of samples, and $c = \frac{3}{8}$.

4. **Feature Selection**: Features are selected using a correlation threshold, excluding features with a correlation above 0.9 with other features.

## D DISCUSSION

### D.1 FEATURE SPACE PROCESSING IN BIOLOGICAL REPRESENTATION LEARNING

Our approach to representation learning in Cell Painting data distinguishes itself through a deliberate focus on engineered feature space processing. Unlike approaches that directly operate on raw image data, our method leverages CellProfiler-generated features, offering several critical advantages:

- **Biological Interpretability:** By maintaining direct mapping to established cellular measurements, we ensure that learned representations retain meaningful biological context. Each transformed feature can be traced back to its original morphological or biochemical interpretation, a crucial requirement for biological research.

- **Methodological Innovation:** Our approach bridges traditional feature-based analysis with modern representation learning techniques. We demonstrate that sophisticated machine learning transformations can be achieved while preserving the semantic meaning of individual cellular features.

- **Practical Workflow Integration:** The method supports immediate adoption within existing Cell Painting protocols, addressing a critical need in drug discovery and biological research workflows.

### D.2 BATCH CORRECTION AND REPRESENTATION LEARNING CHALLENGES

Batch correction in biological data presents unique challenges that our approach systematically addresses:

1. Removing technical variations while preserving biological signals

2. Maintaining interpretability of cellular measurements

3. Enabling cross-experimental comparability

4. Handling high-dimensional, complex biological datasets

Our three-stage training process specifically targets these challenges by learning to distinguish batch-specific artifacts from biologically meaningful features.

### D.3 Limitations and Model Generalizability

The current implementation of CellPainTR introduces both significant advances and notable limitations:

**Current Constraints:**

- Fixed pretrained source context tokens limit direct processing of unseen data sources
- Computational intensity of the training process
- Reliance on CellProfiler-generated features

**Adaptation Strategies:** We propose a straightforward extension mechanism for handling new data sources:

1. Extend the source embedding layer
2. Fine-tune new source context embeddings
3. Follow the established training procedure

### D.4 Future Research Directions

The methodological framework we introduce opens several promising avenues for future research:

- Developing more dynamic source context token adaptation mechanisms
- Exploring end-to-end learning approaches for cellular imaging
- Expanding applicability across diverse biological datasets
- Investigating transfer learning strategies for cellular phenotype representation

### D.5 Broader Impact

Beyond technical innovations, our work contributes to a broader scientific objective: making advanced machine learning techniques more accessible and meaningful in biological research. By maintaining interpretability and providing a robust methodological framework, we offer a critical stepping stone for more transparent and impactful computational approaches in cellular analysis.

The approach demonstrates that sophisticated representation learning can be achieved within constrained, interpretable feature spaces, challenging the prevailing notion that advanced machine learning requires complete abstraction from domain-specific measurements.

### D.6 Practical Considerations for Model Stages and Computational Efficiency

The proposed three-stage approach—CellPainTR(1), CellPainTR(2), and CellPainTR—offers flexibility depending on the requirements of the analysis. Each stage reflects a trade-off between computational demands, batch correction efficacy, and biological signal preservation.

**CellPainTR(1):** This stage serves as a foundational representation learning model, achieving a good starting point for biological signal extraction. However, it retains pronounced batch effects between data sources, making it more suitable for exploratory analyses or scenarios where inter-source batch effects are less critical (Figure 7 6 5, Tab 1).

**CellPainTR(2):** Optimized for single-source datasets, CellPainTR(2) effectively addresses intra-source batch effects while retaining biological signals with high fidelity. This model is particularly useful for studies focused on single-source data, such as analyzing cellular responses within a specific experimental batch or environment (Figure (Figure 7 6 5, Tab 1).

**CellPainTR:** The final stage represents the most balanced model for multi-source datasets, excelling in inter-source batch correction. While there is a minor loss in biological signal fidelity compared to CellPainTR(2), this trade-off enables robust integration and analysis across large-scale, multi-source data, making it well-suited for meta-analyses and cross-experiment studies (Figure 7 6 5, Tab 1).

These distinctions provide practical guidance for researchers on selecting the appropriate model stage based on their experimental setup and goals. The additional visualizations E included in the appendix further illustrate these trade-offs, emphasizing the nuanced improvements achieved by the final stage.

**Computational Efficiency and Reproducibility:** The training times reported in this study reflect an unoptimized setup, involving a single Quadro 4000 GPU and no advanced strategies such as distributed training or mixed precision. The reported times are provided for transparency and to guide users who may wish to replicate the experiments using the current public codebase.

To reduce computational time, several straightforward optimizations can be implemented:

- Utilizing modern GPUs with higher processing power.
- Incorporating distributed training techniques for parallel computation.
- Leveraging mixed precision to reduce memory requirements and training time.

It is important to note that the primary focus of this work is the methodological design and its effectiveness in addressing batch correction and biological signal retention, rather than computational efficiency. Nonetheless, these optimizations can significantly lower the training cost without altering the methodology.

By providing these clarifications, we aim to facilitate the reproducibility of our results while ensuring that researchers can adapt our approach efficiently to their specific computational resources and research goals.

# E    VISUALIZATION

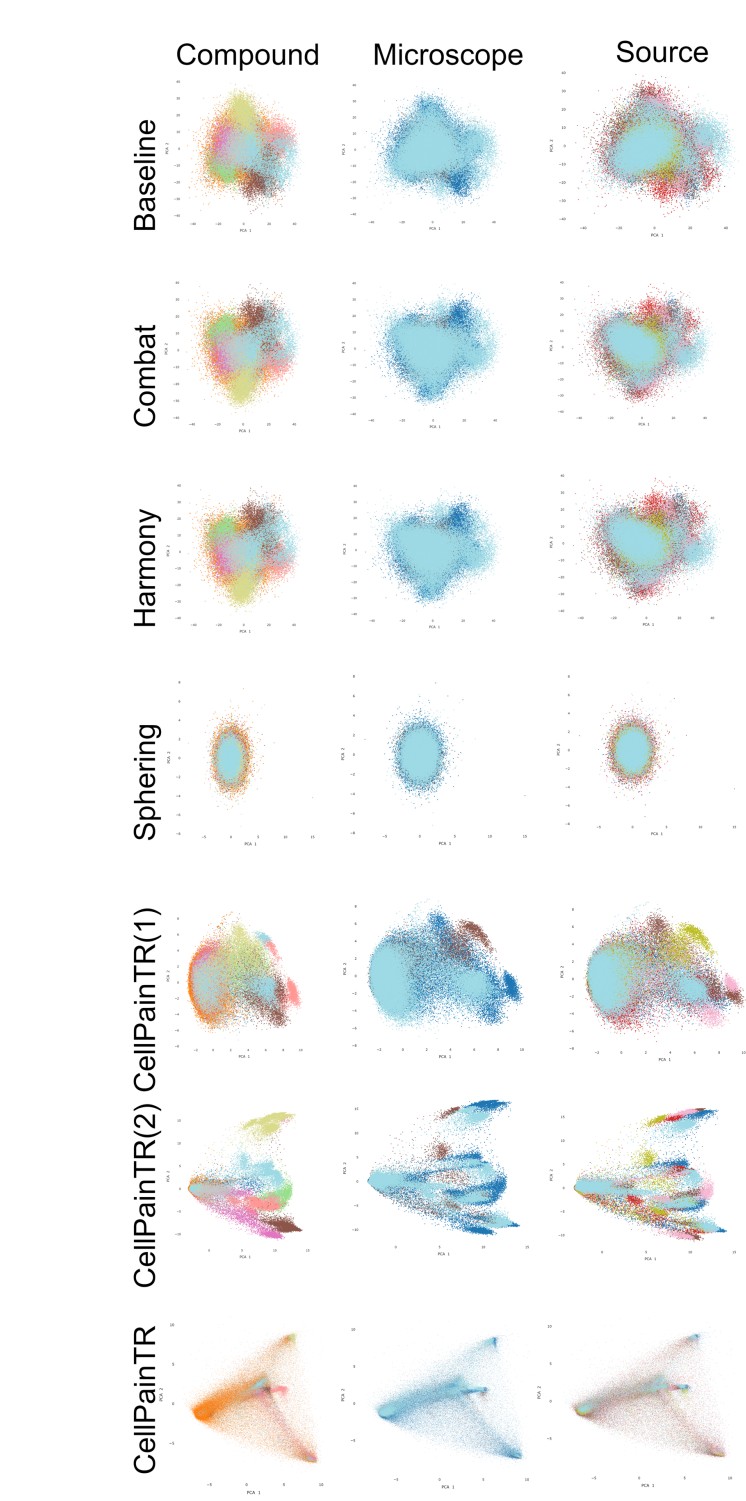

Figure 5: **Complete PCA Comparison**. In this figure, we compare all methods listed in Table 1. Although the dimensionality reduction method used is PCA, we observe clear structures emerging from the CellPainTR(2) method, indicating strong signal preservation and effective optimization within the learned representation.

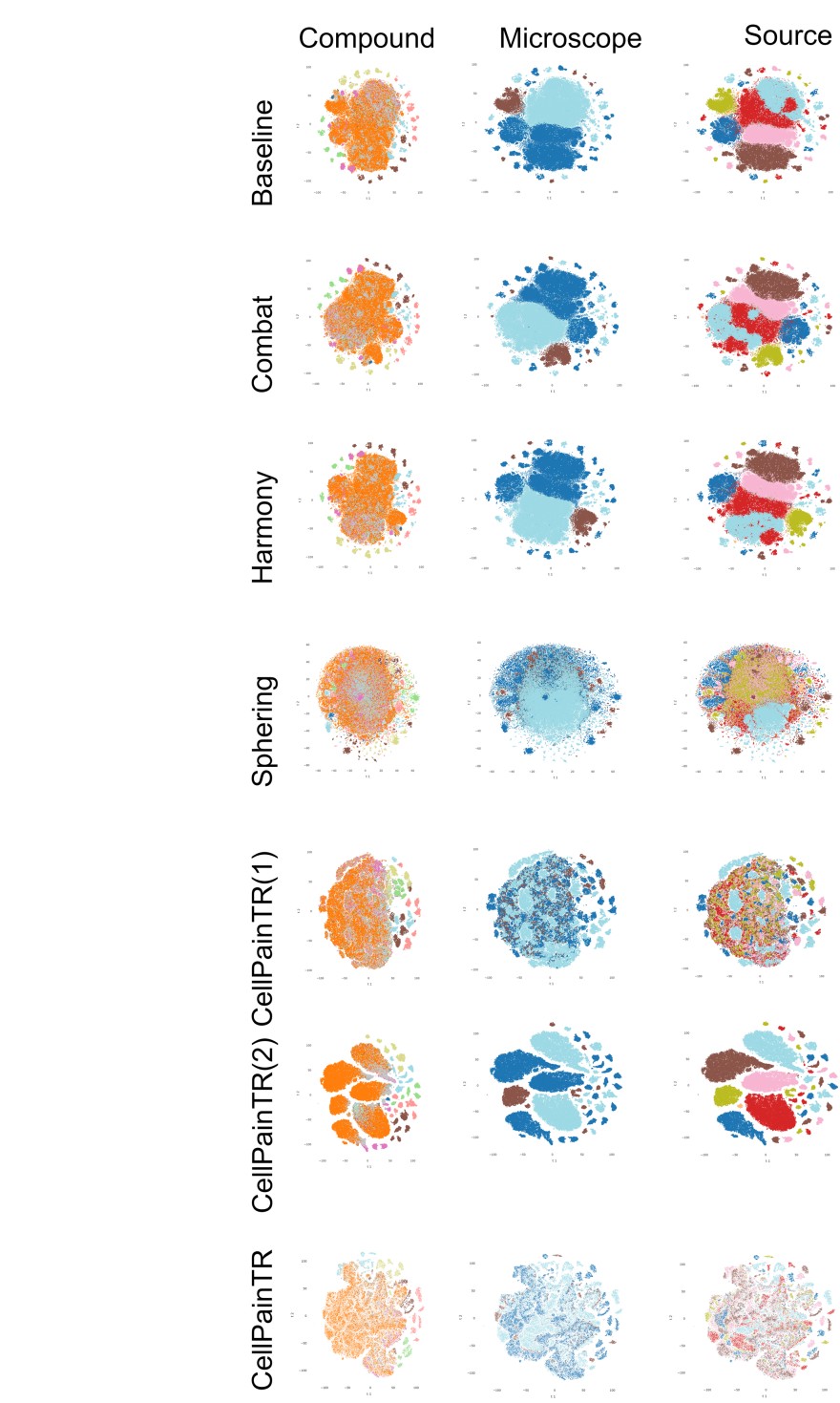

Figure 6: **Complete t-SNE Comparison**. In this figure, we compare all methods listed in Table 1. For all steps of CellPainTR, we observe a better organization in the t-SNE, suggesting strong retention of biological signals. Additionally, for CellPainTR(1) and CellPainTR, the microscope and source batch effects are largely corrected. On the other hand, CellPainTR(2) shows strong fragmentation, which is expected due to the intra-source nature of its objective. Interestingly, CellPainTR(2) organizes its manifold by compound regions. Since CellPainTR(2) does not have access to different source compounds simultaneously, this emergent organization is noteworthy.

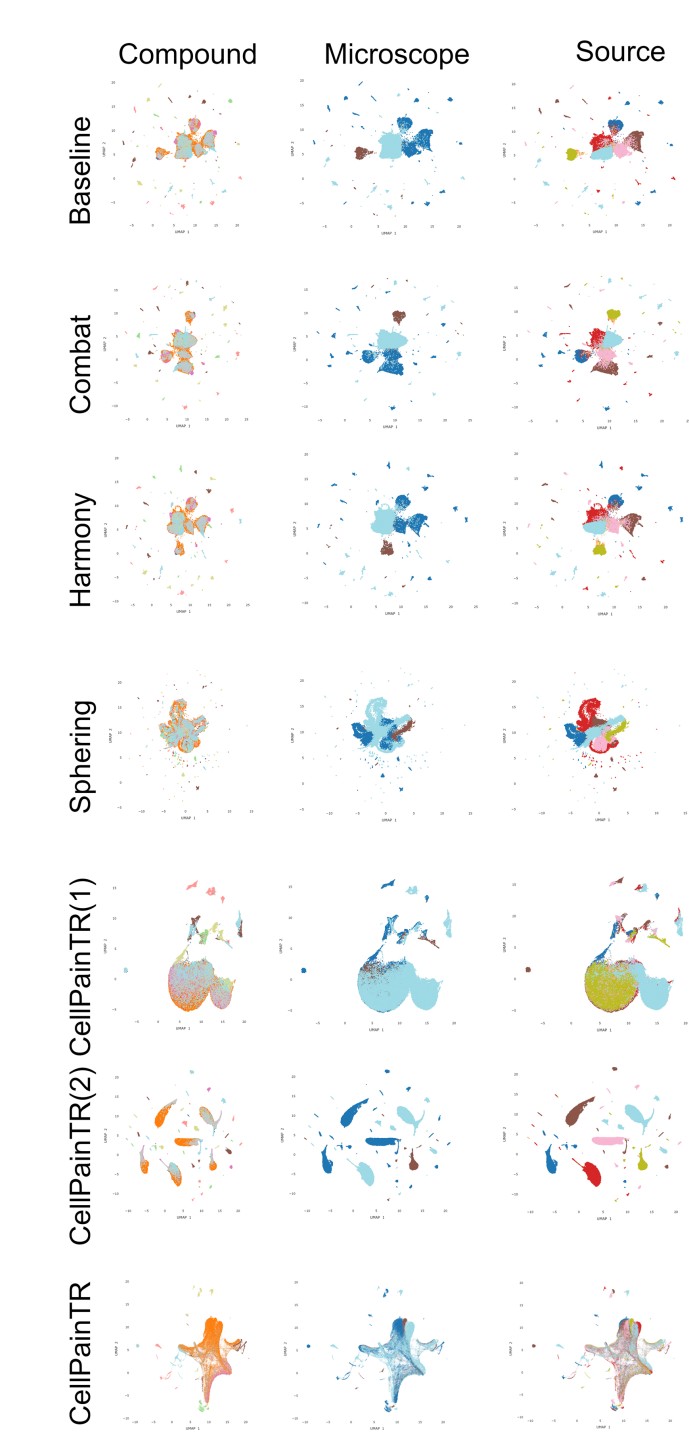

Figure 7: **Complete UMAP Comparison**. In this figure, we compare all methods listed in Table 1. For each of the baseline methods, we observe a strong fragmentation in the visualization, suggesting a pronounced batch effect. In contrast, both CellPainTR(1) and CellPainTR manage this fragmentation effectively. However, due to the training nature of CellPainTR(2), the UMAP also shows significant fragmentation. Despite this, the intra-source clusters are more cohesive compared to the baseline methods, indicating the method's success in reducing batch effects within individual sources.

