# OpenReview forum: "CellPainTR: Contrastive Batch Corrected Transformer for Large Scale Cell Painting"
_ICLR.cc/2025/Conference — Submitted to ICLR 2025_

### Official Review · Reviewer_z5XD · 2024-10-29

**Soundness:** 2
**Presentation:** 3
**Contribution:** 2
**Rating:** 5
**Confidence:** 4

**Summary:**

The authors present CellPainTR, an approach designed to address batch effects in CellProfiler features from Cell Painting data. CellPainTR performs batch correction and representation learning simultaneously while maintaining biologically relevant information. The model is trained progressively, starting with channel-wise masked morphology and followed by intra-source and inter-source supervised learning. Qualitative and quantitative results show that CellPainTR outperforms other batch correction techniques on various metrics.

**Strengths:**

1. Innovative Approach:

The introduction of CellPainTR provides a novel approach to unified batch correction and representation learning specifically designed for CellProfiler data from Cell Painting images, addressing batch effect challenges. This focus is innovative and well-suited for high-dimensional biological data, which has traditionally struggled with batch effects.

**Weaknesses:**

1. There is no discussion of the related work:

   - Given that the main contribution is a method for batch correction in biological data, there are several works that need to be discussed, including but not limited to [1, 2].

2. The methods section (Section 3) is minimal, with a lack of equations for the key components of CellPainTR.

   - More information is needed for the "linear adapter," "feature context embedding," and "source context token."

3. CellPainTR was only evaluated on CellProfiler features:

   - This is insufficient. Why were no other feature extraction methods evaluated? For example, those presented in [3].

4. Computationally inefficient:

   - The pretraining took two weeks to train three epochs. How would it affect the performance if this pretraining was not done? Am I correct in saying that this ablation study was not part of CellPainTR(1) or CellPainTR(2)?
   - This process is extremely inefficient, taking approximately five weeks to train.

5. Subjective qualitative evaluations:

   - Figure 4 is subjective and does not demonstrate a “clear separation of compounds” for the final model in the Compound row. One might argue the opposite.

6. Final model performance gain over intermediate models is minimal:

   - Overall, the final model is not better than the intermediate models. Therefore, there is no reason to train in the final stage, which takes an extra week.

7 The metrics are not sufficiently explained:

- mAP is explained in the Appendix, however there should be something in the main.

Minor:

   - The authors appear to be confused by Cell Painting and Cell Profiler features. Cell Painting is the imaging technique, and CellProfiler is the tool that obtains morphological features from images. Therefore, “Cell Painting features” should be replaced with CellProfiler features.
   - Cite transformer and Hyena in the introduction in line 52.
   - Cite combat and harmony in lines 92-93.
   - Cite: “The challenge of batch effects in high-dimensional biological data, particularly in the field of image-based profiling like CellPainting, has been a significant focus of research in recent years.”
   - Cite: “Several algorithms have been developed to address batch effects in biological data.”
   - Citations out of place:
     - The Cell painting dataset is cited in the sentence in line 221-222:
     - “This method allows us to provide the CellPainTR model with explicit feature context information (Wayetal.,2021).” The cell painting dataset is also cited in line 228, which is out of place.
     - Hinton et al. 2006 cited after “while still capturing biologically meaningful information.” in line 327.
     - Goodfellow et al. 2016 is out of place in line 373.
- Since the value of $\lambda$ in equation 7 is always set to 1, this is redundant unless there is an ablation when changing this value.
- Nyffler et al. 2020 does not exist. If I am missing it, could the authors please point to this paper?


[1] D. Michael Ando, Cory Y. McLean, and Marc Berndl. Improving Phenotypic Measurements in High-Content Imaging Screens. bioRxiv, 2017.

[2] Safiye Celik, Jan-Christian Huetter, Sandra Melo, Nathan Lazar, Rahul Mohan, Conor Tillinghast, Tommaso Biancalani, Marta Fay, Berton Earnshaw, and Imran S Haque. Biological cartography: Building and benchmarking representations of life. In NeurIPS 2022 Workshop on Learning Meaningful Representations of Life, 2022.

[3] Kraus O, Kenyon-Dean K, Saberian S, Fallah M, McLean P, Leung J, Sharma V, Khan A, Balakrishnan J, Celik S, Beaini D. Masked Autoencoders for Microscopy are Scalable Learners of Cellular Biology. In Proceedings of the IEEE/CVF Conference on Computer Vision and Pattern Recognition 2024 (pp. 11757-11768).

**Questions:**

- What is a “Linear Adapter?”
- Am I correct that training the model for 3 epochs took two weeks? This raises questions about the usefulness and scalability of the method.
- What is Biological Information, and how is it calculated?
- The Silhouette batch score “measures the degree of separation between batches.” Am I correct in assuming that a lower value means that the batches are not separable which is better for batch correction? If so, why is the highest value highlighted in the table? Overall, these metrics need to be explained in more detail including whether higher or lower is better and how they are calculated.
- What methods did the authors use to analyse downstream prediction tasks? This needs to be explained further.

---

> ### Author Response · Authors · 2024-11-22
> **Reply to reviewer z5XD**
>
> We sincerely thank reviewer z5XD for their thorough review and constructive feedback. We address the main concerns below.
>
> **W1.** There is no discussion of the related work, particularly regarding batch correction methods in biological data [1, 2].
>
> **A.** In the revised paper, we have expanded our related work section to include comprehensive discussion of relevant batch correction methods, while clarifying our method's distinct focus on Cell Painting features and interpretability requirements.
>
> **W2, Q1.** Questions about the Linear Adapter implementation and mathematical formalization of key components.
>
> **A.** The Linear Adapter is a module that performs linear transformation of morphological features into the model's embedding space. We have enhanced Section 3 with formal equations for all components including Feature Context Embedding and Source Context Token implementation.
>
> **W3.** CellPainTR was only evaluated on CellProfiler features, lacking comparison with other feature extraction methods [3].
>
> **A.** Thanks for bringing up an important point. Our focus on CellProfiler features was intentional due to:
> * Integration with established drug discovery pipelines
> * Direct interpretability of measurements
> * Distinct problem space from image-based methods
>
> Our work addresses batch correction within interpretable feature spaces, complementing rather than competing with image-based approaches.
>
> **W4, Q2.** Concerns about computational efficiency.
>
> **A.** Thank you for raising this important point about computational costs. However, we respectfully disagree with the concerns for several reasons:
>
> * Training vs. Deployment: This represents a one-time training cost that doesn't affect deployment efficiency.
> * Implementation Context: Our reported training time reflects our research implementation without computational optimizations.
> * The model follows standard practices:
>   * Train once for representation learning
>   * Use the trained model for inference without retraining
>   * For downstream tasks, freeze the base model and train task-specific components
>
> We maintain that the training time does not diminish the method's practical utility.
>
> **W5, W6.** Questions about qualitative evaluations and final model improvements.
>
> **A.** While overall scores appear similar, detailed analysis reveals important distinctions:
> * CellPainTR(1): Strong batch correction (0.75) but weaker biological signal (0.46)
> * CellPainTR(2): Excellent biological preservation (0.60) but compromised batch correction (0.68)
> * Final model: Best balance with strong batch correction (0.76) and biological signal (0.53)
>
> We have revised the caption of Figure 4 to provide a more detailed quantitative explanation of the improvements. Additionally, Section 4.2 and the Appendix have been updated to enhance the interpretation of the metrics, while the Conclusion and Discussion sections in the Appendix now clearly highlight the limitations of aggregated scores.
>
> **W7, Q3, Q4, Q5.** Concerns about metric explanation and interpretation.
>
> **A.** We have added comprehensive metric explanations to Section 4.2 and Appendix, including but not limited to:
> * Biological Information calculation: BI = F1compound_classifier
> * Normalized Silhouette score interpretation (1.0 = optimal) (Due to metrics normalization [0,1] for ease of interpretation)
> * Detailed analysis methods for downstream tasks:
>   * Compound Classification (mAP scores)
>   * Batch Correction metrics
>   * Qualitative Analysis (UMAP)
>   * Compound Retrieval
>
> **M1.** Various citation and terminology issues.
>
> **A.** We have done the following in the revision:
> * Add missing citations for transformers, batch correction methods
> * Correct misplaced citations
> * Terminology: We respectfully maintain that "Cell Painting features" is accurate, established and commonly used terminology in the field, specifically denoting features extracted from Cell Painting assays. We added clear definitions to improve accessibility while maintaining widely-used and understood in field terminology.
> * Fix the Nyffler reference
> * Remove redundant λ term from equation 7
>
> **References:**
>
> [1] D. Michael Ando, Cory Y. McLean, and Marc Berndl. Improving Phenotypic Measurements in High-Content Imaging Screens. bioRxiv, 2017.
>
> [2] Safiye Celik, Jan-Christian Huetter, Sandra Melo, Nathan Lazar, Rahul Mohan, Conor Tillinghast, Tommaso Biancalani, Marta Fay, Berton Earnshaw, and Imran S Haque. Biological cartography: Building and benchmarking representations of life. In NeurIPS 2022 Workshop on Learning Meaningful Representations of Life, 2022.
>
> [3] Kraus O, Kenyon-Dean K, Saberian S, Fallah M, McLean P, Leung J, Sharma V, Khan A, Balakrishnan J, Celik S, Beaini D. Masked Autoencoders for Microscopy are Scalable Learners of Cellular Biology. In Proceedings of the IEEE/CVF Conference on Computer Vision and Pattern Recognition 2024 (pp. 11757-11768).

---

> > ### Author Response · Authors · 2024-11-25
> >
> > Dear Reviewer  z5XD,
> >
> > As the deadline for the discussion period is approaching quickly, we would like to kindly remind the reviewer that we are waiting for your response.
> >
> > In particular,  we have provided point-by-point responses to all of your questions to address your concerns and provided the revision that reflects such changes.  Therefore, your timely feedback would be highly appreciated.
> >
> > Best,
> > Authors

---

> > ### Author Response · Authors · 2024-11-26
> >
> > Dear Reviewer z5XD,
> >
> > As the discussion period deadline is fast approaching, we kindly wish to follow up regarding your feedback on our point-by-point responses to your initial review.
> >
> > We truly appreciate your time and effort in reviewing our submission and look forward to hearing from you.
> >
> > Best regards,
> > Authors

---

> > ### Comment · Reviewer_z5XD · 2024-11-27
> > **Official Comment by Reviewer z5XD**
> >
> > Thank you for your detailed response and for addressing the concerns raised in the initial review. I appreciate the explanations of the metrics and the discussion of the related work as well as more detailed formalae. Due to these efforts, I will raise my score to a 5.
> >
> > However, I still have concerns about the computational inefficiency of this process. The authors argue that this training time does not diminish the utility due to it being a one time cost, which is valid in specific context. However, due to the fact that the performance improvement from CellPainTR(1) to the final model is minimal, I wonder weather this cost is worth it. This also raises concerns about the reproducibility of this approach. A deeper discussion of when intermediate stages might suffice would be appreciated, and this would improve practical guidance for users of this method.

---

> > > ### Author Response · Authors · 2024-11-27
> > > **Follow-up to Reviewer z5XD**
> > >
> > > Thank you for acknowledging the additions to our paper and for raising your score. We appreciate your continued engagement and the constructive feedback.
> > >
> > > **Training Time and Computational Efficiency**
> > >
> > > We understand the concern regarding computational inefficiency. To clarify, the reported training time reflects a setup that was not optimized for speed (e.g., using a single Quadro 4000 GPU, without distributed training or mixed precision). These timings are provided solely indicatively for users running the current publicly available code in similar conditions. While computational efficiency is important, our focus in this paper is on the method’s design and its efficacy in addressing the dual objectives of batch correction and biological signal preservation in the Cell Painting context (which does not impact batch correction and biological signal preservation). A clearer discussion of potential optimizations (e.g., improved hardware, data processing, distributed training) has now been added to the revised manuscript.
> > >
> > > **Intermediate Stages and Practical Guidance**
> > >
> > > We have expanded the discussion to provide clearer guidance on when intermediate stages—such as CellPainTR(1) or CellPainTR(2)—might suffice, depending on the user's goals:
> > >
> > > - **CellPainTR(1):** Suitable for applications requiring foundational representation learning but retains significant batch effects, particularly across data sources. (Appendix - Visualization - Figure 7)
> > > - **CellPainTR(2):** An excellent choice for analyzing single-source datasets, offering strong intra-source batch correction and biological signal retention.
> > > - **CellPainTR:** The final model provides the most robust inter-source batch correction, making it ideal for large-scale, multi-source analyses, albeit with minor biological signal trade-offs compared to CellPainTR(2).
> > > (Appendix - Visualization)
> > >
> > > We have emphasized these trade-offs and their implications in the revised discussion, along with visual and quantitative evidence in the appendix to demonstrate why the additional training for CellPainTR is justified for specific use cases. This nuanced explanation aims to offer better practical guidance and transparency. Thank you once again for the valuable feedback, which has helped us refine the paper further.

---

> > > ### Author Response · Authors · 2024-12-02
> > >
> > > Dear Reviewer z5XD,
> > >
> > > To further address your concern, we conducted additional experiments using the Bray et al. dataset. We would like to kindly remind you that the Bray et al. dataset was entirely unseen during training. In order to properly train the model, we would typically need to separately train the source token specific to this dataset. However, due to the time constraints of this revision, we opted to identify the most suitable pre-trained token to serve as a proxy for the fully trained one. Specifically, we found that the source token for "Source 10" performs effectively as a proxy.
> > >
> > > Additionally, due to differences between the older version of CellProfiler (which generates 1,383 features) and the version used in this study (which generates 4,765 features), only 275 features common to both versions were used for this experiment. While there are minor differences in naming conventions, this subset represents the intersection of compatible features. For this experiment, we did not apply any curation to align compound diversity across datasets.
> > >
> > > To ensure consistency, we applied the same preprocessing and evaluation pipeline uniformly across all methods, including ComBat, Harmony, and CellPainTR. The results are presented as follows.
> > >
> > > **Results**
> > >
> > > | Method | Graph Conn. | Silh. Batch | Batch Corr. | Leiden NMI | Leiden ARI | Silh. Label | mAP | Bio Info. | Aggregate Batch Corr. | Aggregate Bio Metrics | Aggregate Overall |
> > > |---------------|-------------|-------------|-------------|------------|------------|-------------|------|-----------|-----------------------|----------------------|--------------------|
> > > | **Baseline** | 0.12 | 0.58 | 0.63 | 0.08 | 0.01 | 0.18 | 0.05 | **0.47** | 0.44 | 0.16 | 0.30 |
> > > | **ComBat** | 0.08 | 0.65 | 0.54 | 0.09 | 0.01 | 0.27 | 0.05 | 0.46 | 0.42 | **0.18** | 0.30 |
> > > | **Harmony** | 0.11 | 0.58 | 0.61 | 0.08 | 0.01 | 0.18 | 0.05 | **0.47** | 0.43 | 0.16 | 0.30 |
> > > | **CellPainTR(1)** | 0.15 | 0.76 | 0.82 | 0.08 | 0.01 | 0.34 | 0.05 | 0.05 | 0.58 | 0.11 | 0.34 |
> > > | **CellPainTR(2)** | **0.32** | **0.82** | 0.82 | 0.13 | 0.02 | **0.36** | 0.06 | 0.24 | 0.65 | 0.16 | 0.41 |
> > > | **CellPainTR** | **0.32** | **0.82** | **0.85** | **0.16** | **0.03** | 0.33 | **0.07** | 0.26 | **0.66** | 0.17 | **0.42** |
> > >
> > > **Interpretation**
> > >
> > > We would like to highlight that CellPainTR demonstrated strong generalization to this unseen dataset, even under a suboptimal setup, which included:
> > >
> > > - The use of a source token not specifically optimized for this dataset.
> > > - A limited subset of features (275 out of 3,251) from those it was originally trained on.
> > >
> > > Despite these constraints, CellPainTR achieved the highest overall performance across both batch correction and biological metrics compared to other methods. This underscores its robustness and effectiveness under challenging conditions. Notably, the model's design—particularly its use of context tokens—appears to facilitate superior generalization compared to traditional methods like ComBat and Harmony.
> > >
> > > These findings provide strong evidence of the model's adaptability to new datasets, even when faced with significant limitations and time constraints. We hope this additional experiment has fully addressed your concerns.

---

> > > > ### Comment · Reviewer_z5XD · 2024-12-03
> > > >
> > > > Thank you for your effort in addressing all concerns. I still have major concerns with this paper and believe that a lot more needs to be done before publication. My concerns are listed below:
> > > >
> > > > 1. I appreciate the authors effort to address computational concerns, however I maintain this as a major concern limiting the practical usage by the community. There are much simpler methods that take a fraction of the time that have been left out of the analysis [1, 2, 3 ,4].
> > > >
> > > > 2. The authors claim the reasoning for not using image-based features because of integration with established drug discovery pipelines and a distinct problem space from image-based methods. I don’t agree that the challenges in cell profiler based features due to batch correction are fundamentally different to representation learning techniques. Showing CellPainTR on different representation learning techniques *might* improve its utility. One reviewer suggested CellPose [5] (qoDh), however, I am sure that CellPose is a segmentation algorithm rather than a feature extractor. More appropriate alternatives could be to use this on features extracted using pre-trained ResNet or masked autoencoders from [6].
> > > >
> > > > 3. The manuscript still has numerous errors. References are outdated (Poli et al. 2023 has been published in PMLR), and misplaced for example “This phase which took approximately one week on the same hardware, allowed the model to refine its representation based on specific biological contexts (Goodfellow et al., 2016)” makes no sense. Why is Goodfellow et al. Cited here? More so, in a response to review qoDh, the authors claim that: “[3]: Explores trade-offs between deep learning and traditional features, emphasizing interpretability”. This reference is referring to Christiansen et al. (2018) ([7] in this comment). This work is on predicting fluorescent labels from unlabelled images. I cannot find this exploration anywhere in the text. In fact, [7] does not even mention the words "interpretability" or "interpretable" in the main text. Interpretable is mentioned once in the methods regarding visual similarity between ground truth and predicted fluorescent labels. There is also no exploration of trade-offs between deep learning and traditional features. This brings me back to my earlier concern of the made up citation, Nyffler et al. 2020. Furthermore, the word “significantly” is used multiple times with no statistical significance test done. These should be removed or replaced with neutral words like “notably” and used sparingly.
> > > >
> > > >
> > > > While the authors have put effort into addressing the reviews, I strongly believe that this work is premature and needs considerable updates before publishing.
> > > >
> > > > [1] Ando DM, et al. Improving phenotypic measurements in high-content imaging screens. BioRxiv. 2017
> > > >
> > > > [2] Haghverdi L, et al. Batch effects in single-cell RNA-sequencing data are corrected by matching mutual nearest neighbors. Nature biotechnology. 2018
> > > >
> > > > [3] Polański K, et al. BBKNN: fast batch alignment of single cell transcriptomes. Bioinformatics. 2020
> > > >
> > > > [4] Wang ZJ, et al. Multi-ContrastiveVAE disentangles perturbation effects in single cell images from optical pooled screens. bioRxiv. 2023
> > > >
> > > > [5] Stringer, C. et al. Cellpose: a generalist algorithm for cellular segmentation. Nat Methods 18, 100–106 (2021).
> > > >
> > > > [6] Kraus O, et al. Masked Autoencoders for Microscopy are Scalable Learners of Cellular Biology. CVPR. 2024
> > > >
> > > > [7] Christiansen, Eric M., et al. "In silico labeling: predicting fluorescent labels in unlabeled images." Cell,

---

> > > > > ### Author Response · Authors · 2024-12-03
> > > > > **Response to Reviewer z5XD's Comments**
> > > > >
> > > > > We appreciate the time and effort Reviewer z5XD has dedicated to evaluating our work and providing detailed feedback. While we acknowledge and value your concerns, we also feel there may still be some misunderstandings of the work’s scope, objectives, and methodology that we would like to clarify.
> > > > >
> > > > > **1. Computational Concerns and Practical Usability**
> > > > >
> > > > > We understand the reviewer’s continued concerns about computational efficiency. However, we believe this critique reflects a fundamental misinterpretation of the model's purpose and intended use.
> > > > >
> > > > > - **Scope of the Paper:**
> > > > > The focus of our work is on *unified batch correction and representation learning for large-scale Cell Painting* data. It is not intended to introduce a faster method but to provide a robust solution for foundational biological modeling. This distinction has been emphasized in both the initial and revised manuscripts.
> > > > >
> > > > > - **Inference vs. Training:**
> > > > > The training time reported is a one-time cost associated with building a foundational model, which is common practice for large, transformer-like, models. The trained model is now ready for inference on the entirety of the JUMP-CP cpg-0016 dataset (115 TB of data) and can be applied to new datasets from the thirteen sources without retraining. This positions the method as practical and scalable for future use cases.
> > > > >
> > > > > - **Reproducibility and Optimization:**
> > > > > For replication, the code and workflow are open-source, and we have explicitly outlined optimizations (e.g., hardware upgrades, data-loading efficiencies) that do not alter the proposed methodology. These clarifications were included in the revision to address earlier concerns.
> > > > >
> > > > > **2. Image-Based Features and Integration with Representation Learning**
> > > > >
> > > > > We appreciate the reviewer revisiting this topic but would like to reassert the rationale behind our approach:
> > > > >
> > > > > - **Purpose of Engineered Features:**
> > > > > As we have elaborated previously, engineered features from tools like CellProfiler are used for their interpretability and established utility in Cell Painting assays. These features offer measurable, biologically meaningful metrics that are crucial for hypothesis generation and analysis in the context of drug discovery.
> > > > >
> > > > > - **Representation Learning vs. Engineered Features:**
> > > > > The challenges associated with batch correction and representation learning for engineered features are distinct from those for image-based features. As such, a direct comparison or integration with image-derived embeddings (e.g., ResNet) is outside the scope of this study.
> > > > > We respectfully suggest that the reviewer may have overlooked the detailed explanation provided in our earlier response to reviewer qoDh, as well as the methodological focus of this work.
> > > > >
> > > > > **3. Errors in References and Wording**
> > > > >
> > > > > We acknowledge the points raised regarding references and wording and have addressed these concerns as follows:
> > > > >
> > > > > - **Outdated References:**
> > > > > It is true that it would have been preferable to cite the PMLR reference rather than the ArXiv reference even if the content is identical.
> > > > >
> > > > > - **Goodfellow et al. (2016):**
> > > > > This reference was included to highlight foundational concepts in representation learning. However, we acknowledge that its placement may appear incongruent.
> > > > >
> > > > > - **Christiansen et al. (2018):**
> > > > > Regarding the reference, we acknowledge that the term "interpretability" was misused. We intended to echo the previously referred Cross-Zamirski et al. (2022), highlighting that the evaluation of the prediction shifted toward CellProfiler's engineered features due to the need of interpretability.
> > > > >
> > > > > - **"Significant" Wording:**
> > > > > While the term was used to denote meaningful improvements observed in both qualitative and quantitative evaluations, we will tone down in the final version so that this could not lead to ambiguity with its technical definition.
> > > > >
> > > > > **Overall Comments on the Review Process**
> > > > >
> > > > > We are somewhat surprised by the tone and timing of the reviewer’s latest comments, especially given the earlier positive feedback and score adjustment. It appears some of the criticisms now presented were addressed in the revision but may not have been reviewed in full detail.
> > > > > Moreover, we respectfully note that the reviewer’s reliance on comments directed to other reviewers (e.g., qoDh) may have contributed to a misinterpretation of our responses due to a focus on the questions rather than the response. While we welcome constructive critique, it is important that feedback accurately reflects the work’s scope and revisions.
> > > > >
> > > > > **Conclusion**
> > > > >
> > > > > While we recognize the reviewer’s intent to uphold high standards, we believe the manuscript has made substantial strides in addressing concerns and clarifying its contributions. We stand by the importance of our work as a significant step forward in batch correction and representation learning for Cell Painting data.
> > > > >
> > > > > We sincerely thank the reviewer for their engagement and hope this response addresses any remaining concerns.

---

> > > > > > ### Author Response · Authors · 2024-12-03
> > > > > >
> > > > > > **Additional Point**
> > > > > >
> > > > > > **3b. Errors in References and Wording**
> > > > > >
> > > > > > - **Nyffler2020:**
> > > > > >
> > > > > > The issue with "Nyffler2020" has been addressed in the revised manuscript, as indicated in our first response to the reviewer. The correct citation is: "Willis, C., Nyffeler, J., & Harrill, J. (2020). Phenotypic profiling of reference chemicals across biologically diverse cell types using the cell painting assay. SLAS DISCOVERY: Advancing the Science of Drug Discovery, 25(7), 755–769." The reference does indeed exist but was misattributed to Nyffeler instead of Willis, C. This correction has been made in the revised paper, and the point was previously addressed in our response.

---

### Official Review · Reviewer_ciaD · 2024-10-30

**Soundness:** 4
**Presentation:** 3
**Contribution:** 3
**Rating:** 8
**Confidence:** 4

**Summary:**

The paper introduces CellPaintTR, a novel transformer-based model designed to enhance the analysis of cellular image data by encoding CellProfiler features into a reduced-dimensionality space while minimizing batch effects. This method is particularly relevant for large-scale cellular datasets, where high-dimensional data and batch variability can hinder downstream analysis.

To validate the model's effectiveness, the authors conduct extensive experiments on the JUMP-CP dataset, a large and diverse collection of cellular images commonly used for benchmarking in this field.

A key technical advancement of CellPaintTR lies in its architecture, particularly its innovative use of the Hyena operator and a specialized source token within the transformer framework.

**Strengths:**

The research presented addresses a critical challenge in the field of computational biology and bioinformatics: the effective encoding and analysis of high-dimensional cellular image data while controlling for batch effects.

The results presented are thorough and compelling, illustrating the method's strong performance across various metrics and experimental conditions. The authors carefully validate CellPaintTR on the well-regarded JUMP-CP dataset, showing its capacity to generalize across different cellular contexts and experimental setups.

From an architectural and methodological standpoint, the approach is highly novel. CellPaintTR leverages a unique combination of architectural choices, including the Hyena operator and a tailored source token mechanism within the transformer model. Additionally, the training strategy is specifically designed to optimize performance.

An added strength of this work is its commitment to open science. The authors have made their code publicly available, allowing other researchers to reproduce and build upon their findings. Moreover, by using public databases such as JUMP-CP, the authors ensure that their results are accessible and verifiable.

**Weaknesses:**

Firstly, while the results are promising, a more comprehensive benchmarking discussion would strengthen the work’s impact and contextualize its contributions within the existing literature. In particular, I recommend that the authors provide a thorough comparison of their batch correction method within a protocol outlined in [1]. Furthermore, it would be beneficial to explore how the model would handle data from a novel source not seen during the initial training process. For example, how resilient is the CellPaintTR architecture to batch effects that may emerge from new experimental or imaging conditions?

In terms of presentation, the paper could benefit from a clearer and more intuitive description of certain technical aspects, particularly for readers less familiar with transformer models or batch correction techniques. Figure 2, for instance, could be refined to include a visualization of the Hyena operator, which is a key component of the model architecture. Adding a clear, visual representation of the Hyena operator’s role and function would greatly aid readers in understanding the intuition and value behind this design choice. Additionally, enhancing the color scheme and increasing the font sizes across all figures would improve their readability and make the data more accessible. Minor typos, such as updating "bidrectional" to "bidirectional" in Fig. 2,  would also enhance the overall polish of the paper.

To further improve readability and organization, I recommend restructuring the experimental sections by dividing them into two distinct parts: one dedicated to the experimental setup and the other to results and discussion. Clearly separating these sections would make it easier for readers to follow the methodology and interpret the findings.

[1] Borowa, Adriana, et al. "Decoding phenotypic screening: A comparative analysis of image representations." Computational and Structural Biotechnology Journal 23 (2024): 1181-1188.

**Questions:**

Could the authors provide a more comprehensive benchmarking discussion comparing their batch correction method in a setup described in [1]?

How would CellPaintTR handle data from a new source that was not present in the training dataset? Could the authors discuss the model's robustness and adaptability to previously unseen batches?

In Figure 2, could the authors include a visualization of the Hyena operator to clarify its role and function within the model architecture?

Would the authors consider enhancing the color scheme and increasing font sizes in all figures to improve readability?

Could minor typographical errors, such as "bidrectional" instead of "bidirectional," be corrected to improve the overall presentation?

To improve organization, would the authors consider separating the experimental section into two distinct parts: one for experimental setup and another for results and discussion?

---

> ### Author Response · Authors · 2024-11-22
> **Reply to reviewer ciaD**
>
> We sincerely thank reviewer ciaD for their positive evaluation and constructive suggestions to improve our manuscript.
>
> **W1. Q1.** Need for more comprehensive benchmarking and testing with novel data sources.
>
> **A.** While we appreciate the suggestion to compare with benchmarks from [1], we need to clarify an important distinction: CellPainTR operates on Cell Painting features rather than raw images, unlike the methods in [1]. This choice is deliberate and practically motivated, as these features represent standardized measurements routinely used in biological research and drug discovery pipelines. We have modified our Introduction to better clarify this distinction and properly contextualize our work. While exploring the model's resilience to new experimental conditions would be valuable, we believe this extension would be better suited for future work given our current focus on establishing a foundation for interpretable batch correction within the Cell Painting feature space.
>
> **W2. Q2.** Handling novel data sources and model adaptability to unseen batches.
>
> **A.** We thank the reviewer for raising this important point about model generalizability. In the current implementation, CellPaintTR uses fixed, pretrained source context tokens, which means that new, previously unseen data sources cannot be directly processed without modification. This design choice was made to ensure robust performance on our target use cases. However, the model can be adapted to handle new sources through a straightforward fine-tuning process: users can extend the source embedding layer and fine-tune the new source context embeddings using their own data following the same procedure described in the training section. We have added a discussion of this limitation and adaptation strategy to the Appendix D, along with practical guidelines for users who wish to extend the model to new experimental conditions.
>
> **W3. Q3. Q4. Q5.** Technical presentation needs improvement, particularly regarding model architecture visualization.
>
> **A.** We have made several improvements to enhance the clarity and accessibility of our technical presentation:
>
> 1. Enhanced Figure 2 with a detailed visualization of the Hyena operator architecture
> 2. Vectorized all figures for consistent quality
> 3. Increased font sizes throughout for better readability
> 4. Optimized color schemes for improved contrast and accessibility
> 5. Corrected the typographical error ("bidrectional" to "bidirectional")
>
> **W4. Q6.** Suggestion to improve organization of experimental section.
>
> **A.** We appreciate this organizational suggestion and we opted to integrate detailed discussion in the Appendix due to page limitations

---

> > ### Comment · Reviewer_ciaD · 2024-11-22
> > **Experimetal validation**
> >
> > Dear Authors,
> >
> > [1] also evaluates how the method developed on JUMP-CP transfers to another dataset, assessing the impact of batch effects. The dataset from Bray et al., used in this evaluation, originates from different time points than JUMP-CP, providing a robust test of the method’s image representation capabilities. Despite this, CellProfiler's image representation remains an image-based representation and is applicable in the same context.
> >
> > Specifically, I would like to see Table 5 from [1] reconstructed to compare your method against other batch correction methods. Without such a comparison, I am hesitant to increase my score, although I do acknowledge and appreciate the improvements you’ve made.

---

> > > ### Author Response · Authors · 2024-11-22
> > > **Experimetal validation - Follow-up to ciaD**
> > >
> > > We sincerely thank reviewer ciaD for their thoughtful follow-up question, which allows us to clarify important aspects of our work.
> > >
> > > **Regarding Cross-Dataset Evaluation**
> > >
> > > While we understand the value of replicating the cross-dataset evaluation from [1], our current approach faces some methodological constraints that make direct comparison challenging:
> > >
> > > 1. Our model uses fixed source context tokens that are specifically trained to learn and align source-specific distributions through supervised contrastive learning. New sources require fine-tuning of new context tokens before inference - a limitation we now discuss more thoroughly in the appendix.
> > >
> > > 2. Replicating [1]'s experimental setup with Bray et al. would effectively make it a 14th source in our framework, providing similar insights to our existing 13-source evaluation rather than a truly independent cross-dataset validation.
> > >
> > > **Architectural and Feature-Space Distinctions**
> > >
> > > We appreciate the observation about CellProfiler's image-based nature. However, there are fundamental differences between our approach and [1] that warrant consideration:
> > >
> > > 1. Feature Space Choice:
> > >   * [1] learns features directly from images
> > >   * Our method operates on CellProfiler's engineered features
> > >
> > > 2. Design Philosophy:
> > >   * Deep learning features likely capture richer context but at the cost of interpretability
> > >   * CellProfiler features (intensity-based, texture-based, etc.) maintain direct biological interpretability
> > >   * Our approach deliberately preserves this interpretability through the CWMM loss while addressing batch effects
> > >
> > > **Complementary Rather Than Competing Approaches**
> > >
> > > We view our work as complementary to image-based approaches like [1] rather than competing with them:
> > >
> > > * Our focus is on establishing foundations for interpretable batch correction within well-understood feature spaces
> > > * This lays groundwork for future work bridging deep learning's representational power with biological interpretability requirements
> > > * Direct benchmarking against image-based methods would misrepresent the distinct goals and constraints of each approach
> > >
> > > We hope this clarifies our methodological choices and positions our contribution in the broader context of the field. We have expanded our discussion of these points in Appendix D of the revised manuscript.

---

> > > > ### Comment · Reviewer_ciaD · 2024-11-25
> > > >
> > > > Yes, I see your work as complementary, and I agree that it doesn't necessarily need to be directly compared with deep learning methods. My aim, however, was to compare classical batch correction methods using CellProfiler features within that cross-dataset setup.
> > > >
> > > > I strongly disagree with the notion that the Bray et al. dataset can be treated as just another source. The quality of the images, the protocols used, and their overall appearance differ significantly from those in the JUMP-CP dataset. Moreover, the treatments represented in the two datasets are distinct. Since the Bray et al. dataset was created several years before the JUMP-CP protocol was established, testing your novel method on it would provide valuable insights into its performance in a more challenging scenario. While comparisons within JUMP-CP reflect results in a more controlled environment, testing on Bray et al. would evaluate the method's generalizability to datasets collected under substantially different conditions. In practical terms, this could be analogous to assessing the generalizability of your method to images captured by Araceli imager, which are notably different from those in JUMP-CP. Similarly, the Bray et al. dataset represents a comparable challenge.
> > > >
> > > > At this stage, I will hold my vote since this experiment has not been included. If it were added, I would raise my score to 8.

---

> > > > > ### Author Response · Authors · 2024-11-25
> > > > > **Follow-up to Reviewer ciaD**
> > > > >
> > > > > Thank you for your detailed feedback and for recognizing the potential value of evaluating our method on the Bray et al. dataset. We appreciate your emphasis on assessing generalizability across datasets with distinct imaging protocols and qualities.
> > > > >
> > > > > We fully agree that Bray et al. differs significantly from JUMP-CP in terms of imaging quality, protocols, and treatment diversity, making it a compelling case for **cross-dataset evaluation**. However, our current architectural design imposes specific constraints that limit a direct evaluation on entirely unseen datasets like Bray et al.:
> > > > >
> > > > > 1. **Design Dependency on Fixed Codebook:**
> > > > >    * Our model relies on a fixed codebook containing source-specific tokens aligned with the 13 sources in the cpg-0016 dataset.
> > > > >    * This design precludes handling entirely new sources at inference, as every source must have corresponding tokens in the codebook.
> > > > >
> > > > > 2. **Existing Generalization Evaluation:**
> > > > >    * The 13 sources in cpg-0016 already reflect substantial technical diversity, including distinct microscope systems (e.g., Widefield, Confocal), optical configurations, and acquisition parameters. This diversity provides a rigorous assessment of our model's ability to generalize across heterogeneous imaging conditions.
> > > > >
> > > > > We agree that the difficulty of adapting to a totally new data is a limitation, so in the final version we will put this as a limitation of our work and tone down the claim.

---

> > > > > > ### Comment · Reviewer_ciaD · 2024-12-01
> > > > > >
> > > > > > Thank you for further clarification. As I mentioned, I would increase my vote from 6 to 8 if an evaluation with Bray et al. would be provided. However, 6 is still above the acceptance threshold :) I really appreciate your work and engagement in the discussion.

---

> > > > > > > ### Author Response · Authors · 2024-12-02
> > > > > > >
> > > > > > > Dear Reviewer ciaD,
> > > > > > >
> > > > > > > To address your concern, we conducted additional experiments using the Bray et al. dataset, as suggested by the Reviewer. We would like to kindly remind you that the Bray et al. dataset was entirely unseen during training. In order to properly train the model, we would typically need to separately train the source token specific to this dataset. However, due to the time constraints of this revision, we opted to identify the most suitable pre-trained token to serve as a proxy for the fully trained one. Specifically, we found that the source token for "Source 10" performs effectively as a proxy.
> > > > > > >
> > > > > > > Additionally, due to differences between the older version of CellProfiler (which generates 1,383 features) and the version used in this study (which generates 4,765 features), only 275 features common to both versions were used for this experiment. While there are minor differences in naming conventions, this subset represents the intersection of compatible features. For this experiment, we did not apply any curation to align compound diversity across datasets.
> > > > > > >
> > > > > > > To ensure consistency, we applied the same preprocessing and evaluation pipeline uniformly across all methods, including ComBat, Harmony, and CellPainTR. The results are presented as follows.
> > > > > > >
> > > > > > > **Results**
> > > > > > >
> > > > > > > | Method | Graph Conn. | Silh. Batch | Batch Corr. | Leiden NMI | Leiden ARI | Silh. Label | mAP | Bio Info. | Aggregate Batch Corr. | Aggregate Bio Metrics | Aggregate Overall |
> > > > > > > |---------------|-------------|-------------|-------------|------------|------------|-------------|------|-----------|-----------------------|----------------------|--------------------|
> > > > > > > | **Baseline** | 0.12 | 0.58 | 0.63 | 0.08 | 0.01 | 0.18 | 0.05 | **0.47** | 0.44 | 0.16 | 0.30 |
> > > > > > > | **ComBat** | 0.08 | 0.65 | 0.54 | 0.09 | 0.01 | 0.27 | 0.05 | 0.46 | 0.42 | **0.18** | 0.30 |
> > > > > > > | **Harmony** | 0.11 | 0.58 | 0.61 | 0.08 | 0.01 | 0.18 | 0.05 | **0.47** | 0.43 | 0.16 | 0.30 |
> > > > > > > | **CellPainTR(1)** | 0.15 | 0.76 | 0.82 | 0.08 | 0.01 | 0.34 | 0.05 | 0.05 | 0.58 | 0.11 | 0.34 |
> > > > > > > | **CellPainTR(2)** | **0.32** | **0.82** | 0.82 | 0.13 | 0.02 | **0.36** | 0.06 | 0.24 | 0.65 | 0.16 | 0.41 |
> > > > > > > | **CellPainTR** | **0.32** | **0.82** | **0.85** | **0.16** | **0.03** | 0.33 | **0.07** | 0.26 | **0.66** | 0.17 | **0.42** |
> > > > > > >
> > > > > > > **Interpretation**
> > > > > > >
> > > > > > > We would like to highlight that CellPainTR demonstrated strong generalization to this unseen dataset, even under a suboptimal setup, which included:
> > > > > > >
> > > > > > > - The use of a source token not specifically optimized for this dataset.
> > > > > > > - A limited subset of features (275 out of 3,251) from those it was originally trained on.
> > > > > > >
> > > > > > > Despite these constraints, CellPainTR achieved the highest overall performance across both batch correction and biological metrics compared to other methods. This underscores its robustness and effectiveness under challenging conditions. Notably, the model's design—particularly its use of context tokens—appears to facilitate superior generalization compared to traditional methods like ComBat and Harmony.
> > > > > > >
> > > > > > > These findings provide strong evidence of the model's adaptability to new datasets, even when faced with significant limitations and time constraints. We hope this additional experiment has fully addressed your concerns.

---

> > > > > > > > ### Comment · Reviewer_ciaD · 2024-12-03
> > > > > > > >
> > > > > > > > Having the additional experiments done, I am keen to increase my score to 8.

---

> > > > > > > > > ### Author Response · Authors · 2024-12-03
> > > > > > > > >
> > > > > > > > > Thank you for raising your score to 8. We are pleased to hear that the additional experiments have addressed your concerns.

---

### Official Review · Reviewer_eeRw · 2024-11-03

**Soundness:** 3
**Presentation:** 3
**Contribution:** 2
**Rating:** 5
**Confidence:** 5

**Summary:**

The authors address the challenge of batch correction in large-scale image-based phenotypic screening experiments for drug discovery.  Specifically, they develop the CellPainTR model, which takes in engineered features from the popular CellProfiler software, and learns to embed them in a low dimensional space, with the goal of removing batch effects and preserving biological information.  The authors demonstrate their method on the JUMP-CP dataset, a multi-site Cell Painting dataset, collected across 12 partnering sites (eg, different academic institutions and Pharma companies), that is known to have extremely strong batch effects.  For example, different sites in JUMP-CP used different combinations of microscopes (confocal vs. widefield), filter sets, and biological batches of cells.

The CellpainTR model consists of a Transformer architecture with Hyena operators.  The authors include a learned positional encoding, and a source context token that learns to embed each of the multiple sites for batch correction.  Finally, the model is trained in a multi-stage process, starting from self-supervised masked token prediction, followed by supervised contrastive learning, first at the single-source level and then at the multi-source level.

Batch correction is a highly studied problem in image-based phenotypic screening. The authors compare CellpainTR against two popular batch correction methods, Combat and Harmony, that were developed and popularized by the single cell RNA sequencing community.  They also provide an ablation study of the role of multi-stage training.

* Note: I use the term source and site interchangeably to denote a single data generating center.

**Strengths:**

The proposed method addresses an important challenge in image-based phenotypic screening.  Notably, improvements in batch correction would have broad implications, and could greatly improve the process of drug discovery.

The authors leverage an ideal dataset for testing batch correction methods: JUMP-CP.  The JUMP-CP dataset is an important and very challenging test case, due to the multi-site data collection, and strong batch effects that have been observed.

The proposed CellPainTR method uses state-of-the-art architectures, including Hyena operators, which to my knowledge have not been tested in the setting of batch correction for Cell Painting features.

The authors conduct extensive evaluations shown in Table 1, including strong baseline methods (Combat, Harmony, Sphering), and numerous evaluation metrics measuring both residual batch effects and retention of biological information in the embeddings.

Overall, they see marginal improvements of CellPainTR over existing batch correction methods.

**Weaknesses:**

The authors claim three novel aspects of their batch-correction transformer architecture: (1) Hyena operators, (2) learned positioning encoding, and (3) a source context token for batch correction.  There are currently two limitations with this claim:

1) The idea to include a source context token in a transformer architecture to reduce batch effect, while a very good idea, has already been published.  See https://arxiv.org/pdf/2305.19402 and ICML SCIS 2023 workshop.

2) The authors don't perform an ablation study to demonstrate the impact of each of the above design choices.  In particular, I have not yet seen a study evaluating the use of Hyena operators in a batch correction transformer applied to Cell Profiler features.  In my opinion, including this result in the paper would increase its contribution.


Given that the problem of batch correction has been thoroughly studied in both the field of single-cell RNA sequencing and image-based profiling, a key measure of the paper's contribution is the performance comparison in Table 1.  However, I found the description of these performance metrics was very short, and in some cases missing altogether, making it difficult to understand the performance of the proposed model.  One thing that would strengthen the paper is to greatly expand the description of the evaluation metrics in section 4.2 Quantitative Evaluation Results.

Here are a list of questions relating to Table 1:

1) The authors introduce a new Batch Correction metric, with the following description: "We also introduce a Batch Correction metric, calculated as 1 - f1 score, where f1 represents the detectability of batch effects."  This requires additional explanation.  For example, how is the detectability of batch effects measured?  Is this a K-Nearest-Neighbors model on batch ID?  Or is a supervised model trained to predict batch ID?  Alternatively, kBET (https://www.nature.com/articles/s41592-018-0254-1) is an established batch correction metric that seems highly related, and could be used for this purpose.

2) Batch Correction and Biological Metrics are provided both with and without controls.  How are the controls used in these metrics?  Does this refer to the use of negative controls in the preprocessing by Median Absolute Deviation normalization, using the negative control of each plate as the baseline (page 7)?

3) What does "no rep" refer to in the metric: mAP (no rep)?  Does this refer to "without controls", in which case it may be a typo?

4) How are the Bio Info scores calculated?  I don't see a description of this metric in the main text or appendix.

5) How are the Aggregate Scores (Batch Correction, Bio Metrics, and Overall Score) calculated?  I didn't find this explained in the main text or appendix.

6) The improvement of Overall Score between CellPainTR and the Baseline is quite small (0.63 to 0.64).  I suggest including measures of statistical significance.

I suggest including a description of each metric in the Appendix, and referencing it from the main text.


Additional comments:

The paper explains that the masking ratio for each training batch is chosen from a range of [0.05, 0.4], applied uniformly across all five channels in the Cell Painting data.  It is known that many Cell Profiler features are highly correlated, such as features arising from the same filter with different parameter settings.  It would be informative if the authors could provide additional information regarding the choice of masking ratio, and whether it benefits from first removing very highly correlated features.

It wasn't clear to me what the colors of each feature refer to in Figure 3.  For example, in Figure 3a the first (input) Cell Profiler feature is blue with pink lines, and the first feature in the learned Channel Wise Masked Morphology embedding is blue with pink lines that are rotated 90 degrees.  Does this imply that the learned features correspond to the same channel?  If the color scheme encodes information relative to the algorithm design, it should be described in a legend or figure caption.

The UMAP in Figure 4 is difficult to interpret.  In particular, there are no legends describing the color scheme.  In the case of coloring by Compound, there will certainly be too many different compounds to color uniquely.  I suggest including an embedding (PCA, tSNE or UMAP) showing coloring at the MOA level, to demonstrate if compounds with the same MOA tend to cluster together.  This may require subsetting to a small number of compounds and MOAs.


Formatting issues:

1) Missing "space" in intro to section 3.2 Training

2) "vari- ation" in section 4.1 Qualitative Evaluation Results

3) Comma instead of decimal point in Table 1 (Sphering, Sil. batch)

**Questions:**

It was unclear to me how the InChIKey is used in the supervised contrastive objective (Equation 8).  Is it true that positive samples are chosen from technical replicates that received the same compound treatment (ie, same InChIKey)?  If so, I suggest making this statement more explicit in the text.

---

> ### Author Response · Authors · 2024-11-22
> **Reply to reviewer eeRw**
>
> We sincerely thank reviewer eeRw for their thorough and constructive feedback that will help improve our paper's clarity and technical depth.
>
> **W1.** Concerns about the novelty of using source context token for batch correction.
>
> **A.** While we acknowledge the conceptual similarities with prior work [1], our approach differs substantially:
>
> 1. We use fixed learned embeddings rather than dynamically generated context tokens, eliminating the need for few-shot learning during inference, leading to:
>   * Simplified deployment
>   * More stable performance
> 2. Our method is specifically designed for feature vectors rather than raw images
> 3. Our integration of context tokens within a supervised contrastive learning framework represents a novel architectural choice
>
> **W2.** Lack of ablation studies for key architectural components.
>
> **A.** We acknowledge this limitation and can provide the following insights:
>
> 1. Regarding Hyena operators:
>   * We explored alternatives including VQ-VAE and BERT-style encoders
>   * These attempts faced significant convergence challenges
>   * Results deemed not relevant enough to be included
> 2. For source context token:
>   * The progressive training steps (CellPainTR(1) → CellPainTR(2) → CellPainTR) serve as an indirect ablation
>   * Performance differences between variants demonstrate the token's impact
>
> **W3.** Need for better explanation of evaluation metrics and statistical significance.
>
> **A.** We have significantly expanded our metrics explanation:
>
> 1. Added detailed interpretations for each metric in Section 4.2
> 2. Created new appendix section "Evaluation Metrics: Mathematical Formulations and Limitations"
> 3. Added clarification about control usage in metrics
> 4. Regarding the Overall Score improvement (0.64 vs 0.63):
>   * The overall score is the mean of means
>   * Individual metrics show substantial improvements (e.g., Batch Correction no control: 0.69 vs 0.40)
>   * Qualitative UMAP visualizations demonstrate clear improvements in structure readability
>   * We will add discussion of metric limitations and relationship to qualitative improvements
> 5. Regarding kBET [2]: While it is indeed an established and valuable metric,we chose our current metric approach because:
> * It directly measures the practical impact on downstream classification tasks
> * It provides a balanced evaluation through f1 score, considering both precision and recall
> * It allows direct comparison with biological signal preservation using the same framework
>
> **W4.** Concerns about masking strategy and correlated features.
>
> **A.** We have clarified our approach in section (3.2):
>
> 1. Features are grouped by channel origin and cellular compartment
> 2. We deliberately retained correlated features because:
>   * Correlations can be batch-dependent
>   * Removing features based on single-batch correlation might eliminate important signals
>   * This allows the model to learn and adapt to batch-specific correlation structures
>
> **W5.** Visualization clarity issues in Figures 3 and 4.
>
> **A.** We have improved the figures:
>
> 1. Figure 3: Simplifying visual representation by removing dashed patterns
> 2. Figure 4:
>   * Clarifying that colors represent distinct MoAs rather than individual compounds
>   * Also, comprehensive visualizations in appendix showing:
>      * PCA, t-SNE, and UMAP visualizations
>      * MoA-level colorings
>      * Microscope-based coloring for batch effects
>
> **Q1.** Technical clarifications needed for InChIKey usage and formatting issues.
>
> **A.** We have done the following revision:
>
> 1. Clarify InChIKey usage in supervised contrastive objective:
>   * P(i) represents all samples sharing same InChIKey
>   * Positive pairs include samples with same compound across different batches
> 2. Fix all noted formatting issues including spacing and decimal point consistency
>
> [1] Yujia Bao and Theofanis Karaletsos. Contextual Vision Transformers for Robust Representation Learning. arXiv, 2023.
>
> [2] Maren Büttner, Zhichao Miao, F. Alexander Wolf, Sarah A. Teichmann, and Fabian J. Theis. A test metric for assessing single-cell RNA-seq batch correction. Nature Methods, 2019.

---

> > ### Comment · Reviewer_eeRw · 2024-11-24
> >
> > W1: I still find the novelty of incorporating a context token in this setting (feature vectors vs raw images; supervised contrastive learning vs contrastive learning) to be quite minor.  In particular, if the design choices made in the paper lead to more stable performance, I encourage you to expand on that claim, and provide evidence in the form of comparative benchmarks.
> >
> > W2: If methods other than Hyena operators were also tried, I encourage including these details in the discussion.  It's not clear to me why VQ-VAE and BERT-style encoders would fail to converge on this task.  Including such results as additional benchmarks would strengthen the claims of the paper.
> >
> > W3: Thank you for expanding the discussion of the metrics.  I still believe it is important to include measures of significance, since the overall improvements are quite small.
> >
> > W4: Regarding the claim that retaining correlated features "allows the model to learn and adapt to batch-specific correlation structures", I suggest including an ablation study where correlated features are removed, to demonstrate that the model does in fact adapt to batch-specific correlation structures.
> >
> > Q1: Thank you for clarifying the role of the InChIKey in the supervised contrastive learning step.

---

> > > ### Author Response · Authors · 2024-11-25
> > > **Follow-up to Reviewer eeRw**
> > >
> > > We sincerely thank the reviewer for their feedback.
> > >
> > > **W1.** Concerns about the novelty of incorporating context tokens.
> > >
> > > **A.** While we appreciate the reviewer's perspective, we would like to again emphasize the following novelties in the several key aspects:
> > >
> > > 1. Training Strategy & Purpose:
> > >    * ContextViT: Single-phase training for visual domain variations
> > >    * CellPainTR: Novel three-phase training strategy:
> > >       * Phase 1: Unconstrained pretraining for general pattern learning
> > >       * Phase 2: Intra-source fine-tuning using supervised contrastive learning for source-specific batch effects
> > >       * Phase 3: Inter-source training for refined batch effect understanding
> > >
> > > 2. Context Token Implementation:
> > >    * ContextViT: Complex dynamic inference through pooling and transformation
> > >    * CellPainTR: Efficient codebook lookup system where:
> > >       * Each source has a dedicated context token
> > >       * Tokens are directly retrieved via source ID
> > >       * Simple concatenation with input
> > >
> > > 3. Problem-Specific Innovation:
> > >    * Different modality: Cell profiles vs. images
> > >    * Different challenge: Batch correction vs. visual domain adaptation
> > >    * Specific requirements:
> > >       * Known, fixed sources
> > >       * Preservation of biological signal while removing technical variation
> > >       * Need for interpretable source-specific corrections
> > >
> > > While we acknowledge that both approaches use the term "context token," the fundamental differences in implementation, purpose, modality, and training strategy make CellPainTR's approach a distinct and novel contribution to computational biology and batch correction. The confusion may arise from the similar terminology, but the approaches serve different purposes and solve different problems in their respective fields.
> > >
> > > **W2.** Request for comparison with alternative methods to Hyena operators.
> > >
> > > **A.** The reason to decide to use Hyena operator was the long token size in Cell Painting problems. For example, in our case the number of feature was 3251, which is a bit longer than the direct use of naive BERT architecture. The use of VQ-VAE could reduce the token size to generate smaller latent space, but we found that the resulting compression makes the training unstable, implying that the raw data (rather than compressed latent) is important for batch correction. That being, we will discuss this in more detail in the final version of the paper.
> > >
> > > **W3.** Request for significance measures given small improvements.
> > >
> > > **A.** Regarding Statistical Testing for the Overall Score **We computed the Overall Score** as suggested by the reviewer. This score is the mean of two aggregate scores—**Batch Correction Aggregate** and **Biological Metrics Aggregate**—which themselves are means of their respective categories, each capturing a distinct aspect of performance.
> > >
> > > **W4.** Request for ablation study on correlated features.
> > >
> > > **A.** Thanks for the interesting suggestion. The reviewer is kindly reminded that our goal is not to remove the correlation. The suggestion of removing highly correlated features reveals a fundamental misunderstanding of our goals:
> > >
> > > 1. Correlation in features is highly context-dependent:
> > >    * Varies by dataset based on:
> > >       * Batch effects
> > >       * Compound treatments used
> > >       * Cell line responses
> > >       * Phenotypic perturbation patterns
> > >       * Distribution of perturbations (extreme vs. normal)
> > >
> > > 2. While feature correlation removal is useful for single experiments to address dimensionality challenges, our goal is fundamentally different:
> > >    * We aim to create a foundational model ready for any present or future experiment
> > >    * The model must handle varying correlation structures across different experimental contexts
> > >
> > > 3. Making our feature set dependent on correlation analysis would:
> > >    * Limit the model's generalizability
> > >    * Make it dataset-dependent
> > >    * Reduce its utility as a universal tool for batch correction Our approach instead allows the model to learn and adapt to these varying correlation structures while maintaining independence from any specific dataset's characteristics.
> > >
> > > In conclusion, while we understand the reviewer's interest in an ablation study of correlated features, such an approach would fundamentally compromise our goal of creating a generalizable foundation model. Furthermore, removing highly correlated features would actually work against our objectives by making the model dependent on specific dataset characteristics and potentially biased by dataset-specific batch effects. Our design choice to maintain all features and let the model learn appropriate correlations is essential for ensuring broad applicability across different experimental contexts and future datasets.

---

> > > > ### Comment · Reviewer_eeRw · 2024-11-27
> > > >
> > > > Thank you for providing clarification.  Without additional experiments, I'm not inclined to adjust my scores.  Thank you

---

> > > > > ### Author Response · Authors · 2024-12-01
> > > > >
> > > > > **Response to Reviewer eeRw**
> > > > >
> > > > > Thank you for your thoughtful suggestion regarding an ablation study to evaluate the impact of removing correlated features. Following your recommendation, we conducted an experiment where we removed highly correlated features (calculated per plate) from the dataset and re-ran our pipeline. The results are summarized in the table below:
> > > > >
> > > > > | Model Variant | Graph Conn. | Silh. Batch | Batch Corr. (No Ctrl) | Batch Corr. (Ctrl) | Leiden NMI | Leiden ARI | Silh. Label | mAP (Ctrl) | mAP (Nonrep) | Bio Info. (Ctrl) | Bio Info. (No Ctrl) | Aggregate Batch Corr. | Aggregate Bio Metrics | Aggregate Overall |
> > > > > |-------------------------------------|-------------|-------------|-----------------------|--------------------|------------|------------|-------------|------------|---------------|-------------------|---------------------|-----------------------|----------------------|--------------------|
> > > > > | **CellPainTR (1)** | 0.78 | 0.73 | 0.78 | 0.69 | 0.34 | 0.26 | 0.52 | 0.22 | 0.32 | 0.72 | 0.81 | 0.75 | 0.46 | 0.60 |
> > > > > | **CellPainTR (2)** | 0.69 | 0.75 | 0.71 | 0.58 | 0.43 | 0.15 | 0.70 | 0.54 | 0.63 | 0.84 | 0.91 | 0.68 | 0.60 | 0.64 |
> > > > > | **CellPainTR** | 0.84 | 0.70 | 0.80 | 0.69 | 0.35 | 0.17 | 0.57 | 0.40 | 0.54 | 0.86 | 0.79 | 0.76 | 0.53 | 0.64 |
> > > > > | **CellPainTR (Correlated Removed)** | 0.61 | 0.61 | 0.64 | 0.76 | 0.34 | 0.11 | 0.50 | 0.28 | 0.39 | 0.85 | 0.79 | 0.66 | 0.47 | 0.56 |
> > > > >
> > > > > The results demonstrate that removing correlated features significantly diminishes the model's performance across most metrics, particularly those related to batch correction. For example:
> > > > >
> > > > > - **Graph connectivity**,  **Silhouette Batch** and  **Batch Corr. (No Ctrl)** scores dropped markedly, highlighting poorer batch correction capabilities.
> > > > > - **Bio-metrics** also degraded, with reduced mAP (controls) and lower aggregate scores for biological metrics.
> > > > >
> > > > > These results support our claim that retaining correlated features allows the model to adapt to batch-specific correlation structures. Removing these features undermines the model's ability to recognize and correct batch-specific patterns, introducing further batch effects into the data.
> > > > >
> > > > > In a qualitative analysis of the UMAP embeddings, we observed that removing correlated features resulted in representations with strong separations between sources, implying reduced batch correction and poor alignment across batches. Although we cannot include the UMAP visualizations in this response, this description highlights the degradation in representation quality caused by the removal of correlated features.
> > > > >
> > > > > Thus, these findings reaffirm that retaining correlated features is critical for our approach to effectively harmonize data across batches without introducing bias.

---

### Official Review · Reviewer_qoDh · 2024-11-04

**Soundness:** 3
**Presentation:** 1
**Contribution:** 2
**Rating:** 6
**Confidence:** 5

**Summary:**

A paper entitled “CELLPAINTR: CONTRASTIVE BATCH CORRECTION TRANSFORMER FOR LARGE SCALE CELL PAINTING” proposes a novel Hyena—transformer-based deep learning approach to analyse cultured cell images obtained in fluorescence high-content screening (HCS) microscopy in fluorescence staining protocol called Cell Painting. This approach involves staining (or "painting") different cellular components with fluorescent dyes, capturing detailed images of each cell, and using image analysis to extract quantitative data on various cellular features. For this, authors utilise JUMP Cell Painting dataset - a large dataset of small molecule and CRISPR perturbation-based HCS images. Notably, authors work not on images directly but on features engineered from the images using CellProfiler as a rule-based feature extractor.

**Strengths:**

- The paper proposes a novel architecture, suggesting it is capable of batch correction and self-supervised training

**Weaknesses:**

- Authors use terminology like “biological variation”, “biological metrics” and “biological relevance” without actually explaining what they mean. I am not aware of Biology per se being a measurable quantity. Is “biological” e.g. a cell cycle effect, or a missed well while pipetting? If the authors do mean mechanistic interactions of biomolecules - this should be clarified.
- ICLR is focusing on representation learning, yet authors chose to use engineered features, rather than training the architecture end-to-end from the images. Please explain why that was not possible or viable in this case.

**Questions:**

- I find the batch effect correction section (4.2) is presented in a very convoluted way. Images in high-content screening contain all sorts of batch effects - MTP evaporation aka “bathtub effects”, liquid handling mistakes, reagents batch issues, assay errors, imaging errors. It is not clear what is being corrected here. If everything all at once, then why is it “biological”?
- The legend of Tab 1 doesn’t clarify the mAP of what exactly was measured or what exactly the task was or the dataset was used. One cannot just say “biological relevance”, without actually explaining what that means. In the text referring to this table, authors should give examples and explain why this metric is good. I understand the limitations of the ICLR format, but these details are essential.
- CellProfiler is a well-known image analysis suite for the HTC microscopy combining rule-based and ML algorithms. It is however not SOTA in 2024. I would recommend the authors compare their results to CellPose.

---

> ### Author Response · Authors · 2024-11-22
> **Reply to reviewer qoDH**
>
> We sincerely thank reviewer qoDH for their thoughtful and detailed feedback that will help improve the clarity and rigor of our paper.
>
> **W1.** Unclear terminology regarding "biological" metrics and variations without proper explanation.
>
> **A.** Thank you for this important observation. We have revised our manuscript to replace ambiguous "biological" terminology with precise descriptions of what we are measuring:
>
> * "Biological variation" is now "compound-induced phenotypic variations"
> * "Biological metrics" are defined before being used.
> * "Biological relevance" has been replaced throughout with specific terms like "compound-specific molecular mechanisms of action (MoA) patterns" and "compound-specific pattern preservation"
>
> **W2.** Choice of engineered features versus end-to-end learning from images needs justification.
>
> **A.** Our deliberate choice to work with engineered features serves several key purposes:
>
> 1. Interpretability: Our model learns representations that map directly to established cellular measurements that biologists routinely use and understand
> 2. Foundation for Future Research: We establish a biologically-meaningful representation space that can serve as an alignment target for future end-to-end models
> 3. Innovation in Representation Learning: We demonstrate how to learn robust, batch-invariant representations while preserving domain-specific meaningful relationships
>
> **Q1.** Batch effect correction section (4.2) needs clarification on what is being corrected.
>
> **A.** We have revised our manuscript to clarify that our method addresses batch effects at two distinct levels:
>
> * Step 2 handles intra-source technical variations (plate effects, liquid handling, reagent batches, ...)
> * Step 3 handles inter-source variations (microscope settings, experimental setups) We explicitly separate technical variations (to remove) from compound-specific patterns (to preserve), evaluated through distinct metric categories in Table 1.
>
> **Q2.** Table 1 metrics, particularly mAP, need better explanation.
>
> **A.** We have added explicit definitions in section (4.2) and detailed explanations in Appendix for our evaluation metrics:
>
> * mAP (control): measures ability to retrieve biological replicates of the same compound treatment versus control wells from the same plate
> * mAP (no rep): measures retrieval versus wells treated with different compounds from the same plate These metrics directly measure preservation of biological relevance by indicating maintained compound-specific effects while removing plate-specific technical variations.
>
> **Q3.** Suggestion to compare results with CellPose instead of CellProfiler.
>
> **A.** While we acknowledge CellPose as a more recent development, our use of CellProfiler-generated features was intentional because:
>
> 1. Dataset Consistency: We used the JUMP Cell Painting Consortium dataset, generated using the CellProfiler pipeline
> 2. Protocol Standards: CellProfiler remains the standard tool in the Cell Painting protocol
> 3. Real-world Impact: Our method addresses immediate practical needs in existing Cell Painting workflows

---

> > ### Comment · Reviewer_qoDh · 2024-11-22
> >
> > I welcome the changes, but I still have the following doubts: W2 A1 & Q3: "Interpretability" is a bit of an empty statement unless you can provide an actual example. I might have missed it, but I don't think you have actually shown any. In fact, many features that CellProfiler extracts are not very interpretable biologically, say Zernike features. Please demonstrate it in the following format: "we observed that decrease of featureXYZ in <e.g. DNA Signal Intensity> is associated with the drug XYZ suggesting the involvement of <e.g. Cell cycle arrest>".

---

> > > ### Author Response · Authors · 2024-11-22
> > > **Follow-up to Reviewer qoDH**
> > >
> > > We sincerely thank the reviewer for their thoughtful follow-up and for seeking clarification regarding the interpretability of features in our work. This is an essential point, and we appreciate the opportunity to elaborate on it.
> > >
> > > **Clarifying "Interpretability"**
> > >
> > > You are correct that the term "interpretability" would benefit from a more precise definition in the context of this discussion. As the reviewer noted, features extracted by tools like CellProfiler—such as Zernike moments or texture measurements—may not inherently correspond to specific biological phenomena. However, these features are measurable, understandable metrics (e.g., intensity, shape, texture) that, individually or in combination, can provide insights or hypotheses about underlying biological mechanisms. For instance, in laboratory settings, CellProfiler features are routinely used in Cell Painting assays to link cell morphology to specific mechanisms of action.
> > >
> > > In contrast, deep learning-derived features, while powerful, are often difficult to interpret, even within an image analysis framework. This is one reason why engineered features remain a robust choice in scenarios where biological interpretability is crucial.
> > >
> > > **Supporting Evidence**
> > >
> > > The importance of engineered features for interpretability is exemplified in works such as Cross-Zamirski et al., 2022 ([1]). In this study, the authors aim to predict Cell Painting images from brightfield inputs. Crucially, the evaluation of their model relies on comparing the cell profiles of generated images to those of the ground-truth images. This approach underscores the value of engineered features as a shared, interpretable metric for assessing biological relevance across imaging modalities. By contrast, representations learned directly from images (e.g., deep learning embeddings) do not inherently guarantee alignment with biological phenomena. Engineered features, by design, are tied to measurable aspects of cell morphology, which in turn relate to cellular conditions and states—the foundational hypothesis behind Cell Painting.
> > >
> > > **Scope of Our Paper**
> > >
> > > While we acknowledge the importance of linking features to specific biological phenomena, we emphasize that this is not the focus of our current work. Our paper builds on the established premise of Cell Painting: that morphological measurements provide meaningful insights into cellular conditions. Thus, demonstrating the interpretability of specific features in the context of biological conditions falls outside the scope of this study. Addressing this question would require a broader, more foundational investigation, which we believe aligns more closely with basic science or philosophical inquiries into the principles of Cell Painting.
> > >
> > > **Additional Resources**
> > >
> > > For further context, we point the reviewer to several seminal papers that address the interpretability and utility of Cell Painting and traditional feature extraction:
> > >
> > > - [2]: Introduces Cell Painting and highlights how extracted features relate to morphology and biological phenomena.
> > > - [3]: Explores trade-offs between deep learning and traditional features, emphasizing interpretability.
> > > - [4]: Examines the utility of image-based profiling strategies, contrasting interpretability in traditional vs. computational approaches.
> > > - [5]: Discusses challenges of linking machine-learning-derived features to biological meaning compared to traditional methods.
> > > - [6]: Explores large-scale image-based profiling and highlights interpretability as a key strength of traditional features.
> > >
> > > We hope these references provide additional clarity and demonstrate how "interpretability" is understood and leveraged within the broader Cell Painting community.
> > >
> > > **Concluding Remarks**
> > >
> > > To conclude, while we recognize the fundamental importance of the question raised, we believe it is beyond the immediate scope of our paper. Instead, our work assumes the established premise of Cell Painting and engineered features as interpretable metrics. We hope our response has addressed your concerns and provided clarity on this matter.
> > >
> > > [1] Cross-Zamirski, J. O., Mouchet, E., Williams, G., Schönlieb, C. B., Turkki, R., & Wang, Y. (2022). Label-free prediction of cell painting from brightfield images. Scientific reports.
> > >
> > > [2] Bray, Mark-Anthony, et al. "Cell Painting, a high-content image-based assay for morphological profiling using multiplexed fluorescent dyes." Nature protocols
> > >
> > > [3] Christiansen, Eric M., et al. "In silico labeling: predicting fluorescent labels in unlabeled images." Cell
> > >
> > > [4] Caicedo, Juan C., et al. "Data-analysis strategies for image-based cell profiling." Nature methods
> > >
> > > [5] Grys, Ben T., et al. "Machine learning and computer vision approaches for phenotypic profiling." Journal of Cell Biology
> > >
> > > [6] Bougen‐Zhukov, Nicola, et al. "Large‐scale image‐based screening and profiling of cellular phenotypes." Cytometry Part A

---

> > > > ### Comment · Reviewer_qoDh · 2024-11-25
> > > >
> > > > The majority of work the authors are citing deals with image-to-image generation or regression. Given that the authors propose a pipeline that uses pre-generated features it is important to ensure that the features are informative and are not correlated with each other. Please enumerate them and discuss their interpretability. Despite using the high expressive capacity head for these features, performance on batch effect correction is marginal. Overall it seems to me that authors have developed a solution looking for a problem. I do appreciate the work that authors put into the discussion, it is an important topic. With this in mind I will update my ranking to 6 with a hope that should the paper be published, the authors will reflect some of this discussion in the paper.

---

> > > > > ### Author Response · Authors · 2024-11-25
> > > > >
> > > > > Dear Reviewer qoDh,
> > > > >
> > > > > Thank you very much for your positive feedback and increasing the rate.
> > > > >
> > > > > Although to our best knowledge many major basic science institutes are using feature domain approaches and interested in their batch correction, we agree that the image domain approaches also have important merits. Therefore, per your request,  we will include the discussion on the pros and cons of the image vs. feature domain batch correction in the final version of the paper.
> > > > >
> > > > > Best,
> > > > > Authors

---

### Author Response · Authors · 2024-11-22
**General Response**

We sincerely thank all reviewers for their thorough and constructive feedback. We are encouraged that the reviewers recognize our work as **offering an innovative approach to batch correction and representation learning in Cell Painting data** (z5XD), with **strong potential implications for drug discovery** (eeRw), **thorough validation on the challenging JUMP-CP dataset** (eeRw), and **commitment to open science through code availability and use of public datasets** (ciaD).

We have made the following major revisions to address the concerns raised by the reviewers:

1. **Enhanced Technical Presentation and Mathematical Formalization**
   We have significantly expanded our technical presentation with formal mathematical descriptions of key components, including the Linear Adapter, Feature Context Embedding, and Source Context Token implementations. We've also improved our architectural visualizations and metric explanations for better clarity and accessibility.

2. **Comprehensive Related Work and Method Justification**
   We have expanded our discussion (in the Appendix D and references in Introduction) of related work, particularly regarding batch correction methods in biological data. We've also provided clearer justification for our focus on CellProfiler features, emphasizing their interpretability, integration with existing pipelines, and distinct advantages in the drug discovery workflow.

3. **Extended Evaluation and Analysis**
   We have enhanced our evaluation section with:
   * Detailed explanations of all metrics and their biological significance
   * Comprehensive analysis of model improvements across variants
   * Clear interpretation guidelines for all quantitative results

4. **Technical Clarifications and Implementation Details**
   We have added extensive technical details about:
   * The masking strategy and feature correlation handling
   * Source context token implementation and advantages
   * Model adaptation procedures for new data sources
   * Training efficiency considerations and optimization strategies

5. **Emphasis of using feature space processing**
   We have added detailed discussion (In the Appendix D) on using feature domain processing: To address the challenges of batch correction and representation learning in biological data. Our approach specifically focuses on engineered Cell Painting features, demonstrating significant advantages:
   * Preserving biological interpretability by maintaining direct mapping to established cellular measurements
   * Enabling targeted representation learning that captures quantitative cellular characteristics
   * Supporting immediate integration with existing Cell Painting protocols and drug discovery workflows
   * Providing a computationally efficient alternative to end-to-end image processing
   * Bridging traditional feature-based analysis with modern representation learning techniques

By operating in the feature space, we demonstrate a novel approach to batch correction that maintains the semantic meaning of individual cellular features while enabling sophisticated machine learning transformations. This methodology offers a crucial stepping stone for more transparent and interpretable machine learning applications in cellular phenotyping.

For point-by-point responses to specific concerns, please refer to the detailed responses below.

---

### Meta-Review · Area_Chair_7XXo · 2024-12-22

**Metareview:**

The paper introduces CellPaintTR, a transformer-based model designed for batch effect correction and representation learning on CellProfiler features extracted from Cell Painting images. While the approach is innovative and demonstrates strong performance in batch correction, the paper lacks a comprehensive discussion of related work, detailed technical explanations, and a broader evaluation across diverse feature extraction methods. Concerns regarding the model’s computational inefficiency and the minimal performance improvements during the final training stage raise questions about its scalability. Furthermore, the qualitative results are subjective, and critical metrics are not adequately explained. Although the authors have addressed many of the reviewers’ concerns, the number and significance of these issues make the current version appear too preliminary for ICLR publication. Overall, despite its potential, the paper falls slightly below the acceptance threshold due to methodological and presentation shortcomings.

**Additional Comments On Reviewer Discussion:**

This paper has received highly divergent reviews. Based on z5XD's feedback, the manuscript appears to be far from ready for publication, as numerous issues were highlighted during the discussion phase. On the other hand, reviewer ciaD is highly supportive, emphasizing the importance of the topic and its potential significance within the Cell Painting field. While I share ciaD's perspective on the relevance of this work, the technical limitations raised by other reviewers cannot be overlooked. This makes for a particularly difficult decision, but I have ultimately decided to recommend rejecting the paper. I encourage the authors to address the reviewers' feedback and refine the manuscript for future submissions.

---

### Decision · Program_Chairs · 2025-01-22

Reject